# Embryonic geometry underlies phenotypic variation in decanalized conditions

**Anqi Huang[1], Jean-François Rupprecht[1,2], Timothy E Saunders[1,3,4]\***

[1]Mechanobiology Institute, National University of Singapore, Singapore, Singapore; [2]CNRS and Turing Center for Living Systems, Centre de Physique Théorique, Aix-Marseille Université, Marseille, France; [3]Department of Biological Sciences, National University of Singapore, Singapore, Singapore; [4]Institute of Molecular and Cell Biology, Proteos, A*Star, Singapore, Singapore

**Abstract** During development, many mutations cause increased variation in phenotypic outcomes, a phenomenon termed decanalization. Phenotypic discordance is often observed in the absence of genetic and environmental variations, but the mechanisms underlying such inter-individual phenotypic discordance remain elusive. Here, using the anterior-posterior (AP) patterning of the *Drosophila* embryo, we identified embryonic geometry as a key factor predetermining patterning outcomes under decanalizing mutations. With the wild-type AP patterning network, we found that AP patterning is robust to variations in embryonic geometry; segmentation gene expression remains reproducible even when the embryo aspect ratio is artificially reduced by more than twofold. In contrast, embryonic geometry is highly predictive of individual patterning defects under decanalized conditions of either increased *bicoid (bcd)* dosage or *bcd* knockout. We showed that the phenotypic discordance can be traced back to variations in the gap gene expression, which is rendered sensitive to the geometry of the embryo under mutations.

**\*For correspondence:**
dbsste@nus.edu.sg

**Competing interests:** The authors declare that no competing interests exist.

## Introduction

The phenomenon of canalization describes the constancy in developmental outcomes between different individuals within a wild-type species growing in their native environments (*Flatt, 2005*; *Hallgrimsson et al., 2019*; *Waddington, 1942*). To better understand how canalized a developmental process is, we need to quantitatively measure the molecular profiles of the developmental regulatory genes in multi-cellular organisms. In *Drosophila,* the highly reproducible body patterning in adult flies originates from both the reproducible setup of the instructive morphogen gradients and the precise downstream transcriptional readouts early in embryogenesis (*Bollenbach et al., 2008*; *Gregor et al., 2007a*; *Petkova et al., 2014*). In particular, the inter-individual variation of the positional information conferred by gene expression can be less than the width of a single cell (*Dubuis et al., 2013*; *Gregor et al., 2007a*); that is these processes are highly canalized. Given the ubiquity of canalization in nature, such highly reproducible developmental processes are likely not exclusive to insect development.

The developmental canalization that we see in contemporary species is the product of evolution, either as the consequence of stabilizing selection (*Gibson and Wagner, 2000*) or the manifestation of the intrinsic properties of the underlying complex gene regulatory networks (*Siegal and Bergman, 2002*). Canalization can break down in individuals subjected to aberrant genetic mutations or extreme environmental conditions (*Imasheva et al., 1997*; *Rutherford and Lindquist, 1998*). Such individuals in decanalized conditions become sensitive to variations in their genetic background and

external environments, which are otherwise neutral to developmental outcomes. This leads to significantly increased inter-individual variation in phenotypic outcomes.

It is important to characterize the sources of variation in order to understand what canalization is actually buffering against. Interestingly, significant phenotypic variation remains in laboratory animals with isogenic genomes, exposed to homogeneous environments (*Gärtner, 1990*). This indicates that under decanalized conditions, other components besides genetic and environmental variation may also cause phenotypic discordance. Previous work has proposed that stochastic expression of redundant genes predicts the developmental outcome of the mutant individuals (*Burga et al., 2011*). However, in many other cases, it remains elusive as to why mutation increases inter-individual phenotypic variation and what alternative components underlie such variation (*Janssens et al., 2013*; *Surkova et al., 2013*).

To identify potential sources of variation that govern phenotypic discordance, we utilized the highly canalized process of early *Drosophila* embryonic AP patterning. We proposed that the inter-individual variation of the embryonic geometry is one of the factors that predetermine individual patterning outcomes under decanalizing mutations. We found that the patterning system in genetically intact embryos has a striking capacity to buffer against variations in embryonic geometry. When we artificially reduced the aspect ratio of the embryos by more than twofold, the relative boundary positions of segmentation genes showed only a minor shift compared to that of the wild type. The preservation of the boundary positions can be, at least partially, attributed to changes in the Bcd gradient profile due to the rounder embryonic shape in the shorter embryos. Importantly, the inter-individual variations of boundary positions remained comparable to that of the wild type. In contrast, when we introduced decanalizing mutations such as increased maternal *bcd* dosage or *bcd* knock-out, the embryonic geometry becomes predictive of patterning outcomes of individuals. In both scenarios, the inter-individual phenotypic discordance can be traced back to variations in gap gene expression patterns. This suggests that under decanalized conditions particular gene interactions are preferentially affected by geometric constraints. Taken together, our study reveals embryo geometry, or more generally the physical domain in which patterning occurs, is a significant source of variation that can account for phenotypic discordance under decanalized conditions.

## Results

### Embryos with dramatically reduced embryonic length still proceed with normal development

Scaling is one of the most astonishing features of embryonic patterning in dipteran insects. The expression boundaries of the early patterning genes reside at conserved relative positions among closely related dipteran species, although they differ dramatically in egg size (*Gregor et al., 2005*). Such inter-species scaling can be attributed to the Bcd gradient of a conserved characteristic length (*Gregor et al., 2008*). Within the *Drosophila* species, individual embryos of different size also exhibit scaled pattern as a consequence of adapted Bcd production and degradation rate (*Cheung et al., 2011*; *Cheung et al., 2014*). We propose here that another macroscopic variable that the patterning system needs to adapt to in order to ensure scaling is the geometry, or shape of the egg. Given a defined embryonic volume, the embryos can vary significantly in their aspect ratio (AR = embryonic length/embryonic width, *Figure 1A*). Since Bcd molecules diffuse mainly at the cortical region of the embryo (*Gregor et al., 2007b*), varying embryonic geometry affects the effective diffusion area, and thus the length and time scales of the morphogen gradient formation (*Grimm et al., 2010*). Therefore, we first examined how altering embryonic geometry affects embryonic patterning.

The geometry of each embryo is predetermined during oogenesis when the follicle cells surrounding the egg chamber transform the developing egg from a sphere to an ellipse (*Haigo and Bilder, 2011*; *He et al., 2010b*). This process is mediated by the planar cell polarity of the follicle cells and the elliptical shape of the embryos remains unchanged throughout embryogenesis. Here, we used maternal ShRNA to knockdown one of the planar cell polarity core components, atypical cadherin Fat2 specifically in the follicle cells (*Horne-Badovinac et al., 2012*), and we henceforth refer to these as *fat2RNAi* embryos. As Fat2 expression is only inhibited within the somatic cells of the egg chamber using *traffic jam* (*tj*) >Gal4 driver, the fertilized eggs produced by this perturbation remain genetically intact. This reduces the embryonic length from 510 ± 17 (s.d.) μm in wild type to

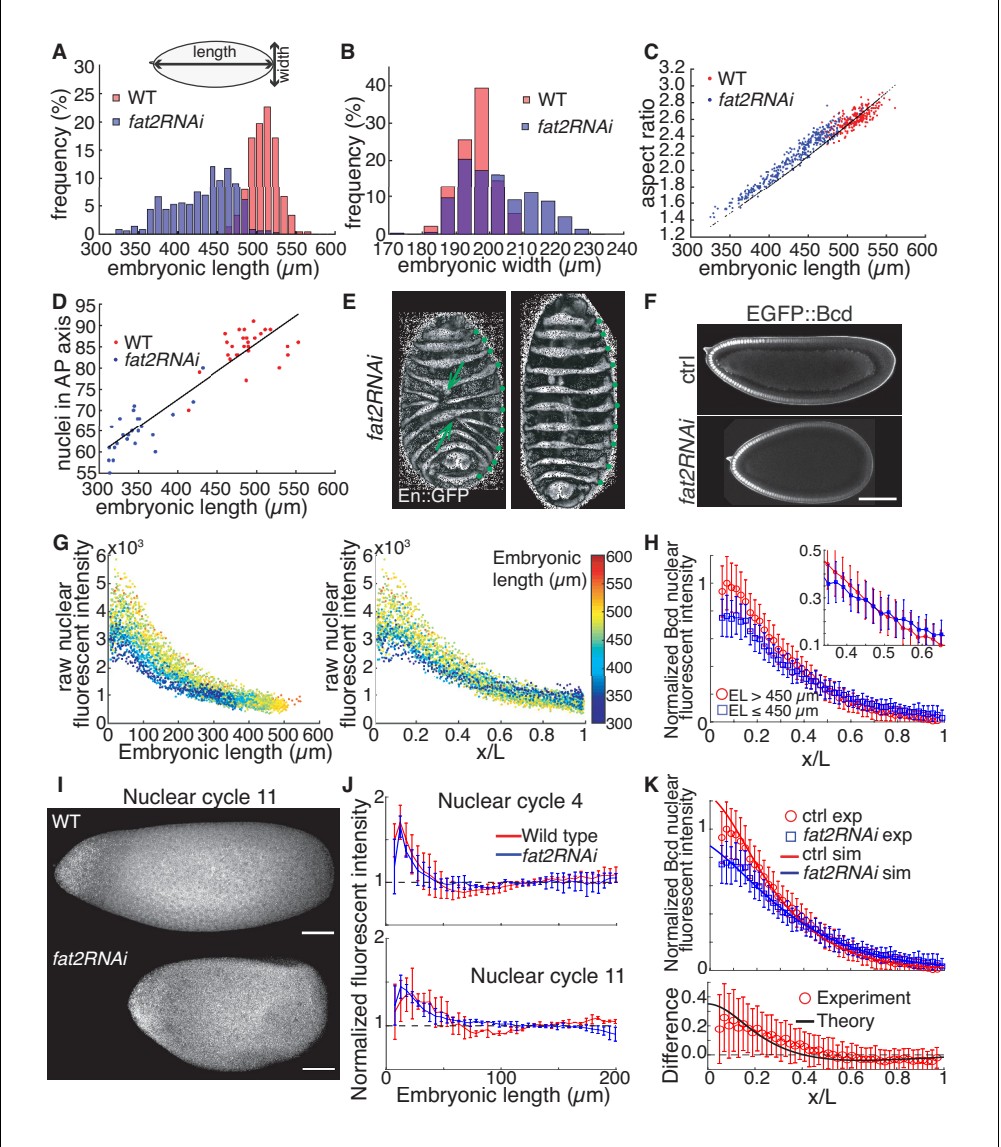

**Figure 1.** The Bcd gradient in embryos with differing geometry is consistent with the SDD model. (**A**) Distribution of embryonic length in wild type (n = 239) and *fat2RNAi* (n = 364) embryos. (**B**) Distribution of embryonic width in wild type (n = 239) and *fat2RNAi* (n = 364) embryos. (**C**) Aspect ratio (EL/EW) against embryonic length for each embryo. Black dots denote expected aspect ratio if embryonic volume is conserved (assuming ellipsoidal geometry). (**D**) Number of nuclei along the AP axis plotted against embryonic length in wild type and *fat2RNAi* embryos. Line indicates linear regression of all data. (**E**) En expression in *fat2RNAi* embryos showing defective dorsal closure (left) or normal morphogenesis (right). Arrows indicate locations of defects. Green dots indicate En stripes. (**F**) Midsagittal plane of ctrl (top) and *fat2RNAi* (bottom) embryos expressing eGFP::Bcd in mid n.c. 14. (**G**) eGFP::Bcd profiles of both ctrl and *fat2RNAi* embryos plotted as a function of absolute distance from the anterior pole (left) or scaled AP position (right). Each dot represents the average concentration in a single nucleus. Colormap indicates the absolute AP length of each individual. (**H**) Mean and standard deviation of nuclear intensity within each 2% EL were computed for group of embryos longer (red, n = 27) and shorter (blue, n = 17) than 450 μm. Inset is close-up of profile near embryo midpoint to show the intersection of the two curves. (**I**) Representative fluorescent in situ hybridization (FISH) against *bcd* mRNA in wild type (top) and *fat2RNAi* (bottom) in n.c. 11 embryos. Scale bar, 50 μm. (**J**) Fluorescent intensity profile of FISH assay along AP axis in n.c. 4 (top) and n.c. 11 (bottom). Normalization to measured fluorescence signal in the region 120 μm from anterior. n = 5, 2 (n.c. 4) and n = 2,2 (n.c. 11) for wild type (red) and *fat2RNAi* (blue) respectively. Error bars show standard deviation. (**K**) Fitting of SDD model to experimentally measured Bcd gradient. All parameters, as outlined in Materials and methods, are kept constant, with only change being embryonic geometry. See

*Figure 1 continued on next page*

*Figure 1 continued*

Materials and methods for details. Lower panel shows intensity difference in experimental measurements and predicted profiles along the AP-axis.

The online version of this article includes the following figure supplement(s) for figure 1:

**Figure supplement 1.** Effects of perturbing embryonic geometry by *fat2RNAi.*

432 ± 40 μm in *fat2RNAi* embryos (*Figure 1A*), with some embryos as short as 320 μm. Meanwhile, the perturbed embryos show an increased embryonic width (EW) along the dorso-ventral (DV) axis (*Figure 1B*, 196 ± 5 μm in wild type and 202 ± 11 μm in *fat2RNAi*). Together, these geometrical variations lead to only a slight reduction of ~8% in the embryonic volume (assuming an ellipsoidal geometry) compared to wild type embryos (*Figure 1C*; *Figure 1—figure supplement 1A–C*). The round eggs are fertilizable and continue with proper embryogenesis (*Figure 1—figure supplement 1D*; *Video 1*).

We examined the nuclear distribution in the blastoderm embryos, as nuclei are the basic units interpreting positional information, and an altered nuclear distribution may affect patterning outcomes. We found that nuclear number along the AP axis decreases proportionally to embryonic length, leaving the inter-nuclear distance unchanged (*Figure 1D*; *Figure 1—figure supplement 1E*). In other words, the number of nuclei to interpret AP positional information reduces from 85 ± 4 in wild type to 65 ± 5 in *fat2RNAi* embryos.

Despite the significant changes in embryonic geometry, the *fat2RNAi* embryos show invariably eleven Engrailed (En) stripes, demarcating the posterior boundary of each body segment including three thoracic (T1 ~T3) and eight abdominal segments (A1 ~A8; *Figure 1E*). We noticed that *fat2RNAi* embryos shorter than 400 μm developed morphological defects in late embryogenesis, where abnormal dorsal closure leads to mismatch between the two lateral sides of the ectoderm (*Figure 1E*, arrows; *Figure 1—figure supplement 1D*). However, such local morphological abnormality is likely due to defective tissue morphogenesis as a consequence of limited physical space, rather than patterning errors (*Video 2*).

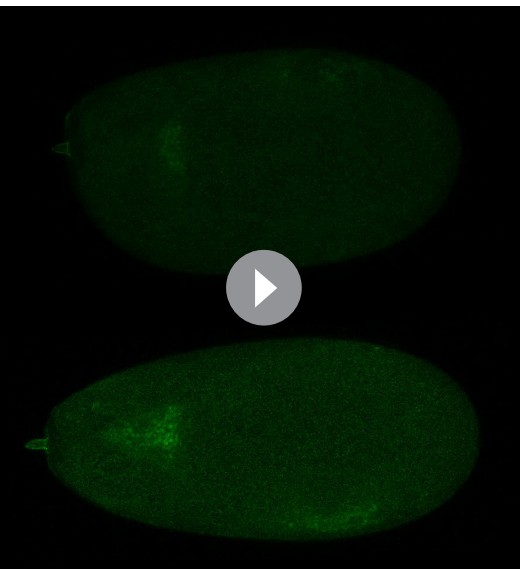

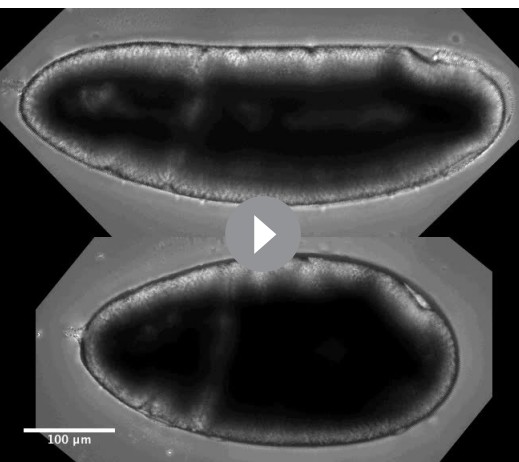

**Video 1.** Embryogenesis of different geometry. Wide field movies of wild type (top) and *fat2RNAi* (bottom) embryos from the onset of gastrulation until hatching.
https://elifesciences.org/articles/47380#video1

**Video 2.** Defective morphogenesis in late embryo development due to extreme embryonic geometry. Confocal movies of *fat2RNAi* embryos expressing *en* >mCD8::GFP from germband retraction to dorsal closure stages. Two embryos in the movie represent individuals with EL shorter (top) and longer (bottom) than 400 μm. The dashed box indicates mismatch between opposing ectodermal tissue during dorsal closure. The numbers indicate the EL of each embryo.
https://elifesciences.org/articles/47380#video2

## Developmental reproducibility is preserved with a minor impact on scaling under geometrical perturbations

Before tackling the consequences of decanalization, we next ask how the AP patterning network, from the initial morphogen gradient formation to gap gene readout, is altered under large geometric perturbations. First, we focus on Bcd, which provides AP positional information in the early embryo. To compare the spatial Bcd concentration profile in different embryonic geometry, we live-imaged eGFP-Bcd fusion protein in both control and *fat2RNAi* backgrounds, and measured the nuclear Bcd intensity along the AP axis at mid nuclear cycle (n.c.) 14 (*Figure 1F*). We found that the absolute Bcd concentration is lower in the anterior half of the embryo in *fat2RNAi* individuals compared to that of control (*Figure 1G and H*) ($p<10^{-3}$, two-sided t-test comparing intensity in region 5–15% EL). Further, Bcd profiles from different individuals intersect near the mid region of the embryo, with shorter embryos showing higher Bcd concentration in the posterior region (*Figure 1G and H*) ($p<10^{-3}$, two-sided t-test comparing intensity in region 65–75% EL).

The formation of the Bcd gradient depends on the amount and the spatial distribution of *bcd* mRNA molecules (*Little et al., 2011*; *Cheung et al., 2014*). To examine *bcd* mRNA distribution under geometrical perturbations, we performed fluorescent in situ hybridization (FISH) against *bcd* mRNA in both wild type and *fat2RNAi* embryos (*Figure 1I*; *Figure 1—figure supplement 1F*). We found that at all stages prior to n.c. 12, the majority of the *bcd* mRNA is localized within the anterior 50 μm of the embryo, regardless of the embryonic geometry (*Figure 1J*). Due to the difficulty in quantifying FISH data, we note that we cannot discount possible variation in the *bcd* mRNA amount between wild type and *fat2RNAi* embryos. However, such variations are likely small and do not substantially alter the overall distribution of *bcd* mRNA. These results are consistent with previous work (*Cheung et al., 2011*), which suggested that embryo volume was a key determinant of *bcd* mRNA levels; in the *fat2RNAi* embryos, volume does not substantially change. Importantly, the geometrical alterations induced in *fat2RNAi* embryos does not change the *bcd* mRNA distribution on an absolute scale.

Given that the *bcd* mRNA distribution does not appear substantially different between the wild type and *fat2RNAi* embryos, what is the cause of the spatial shift in the Bcd profile? We used theoretical modeling to test whether the geometrical changes alone (*Umulis and Othmer, 2012*) are sufficient to explain the altered Bcd profile. The synthesis , diffusion , degradation (SDD) model of Bcd gradient formation provides an excellent description of the Bcd gradient (*Gregor et al., 2007b*; *Durrieu et al., 2018*; *Little et al., 2011*). A one-dimensional description generally works well for modelling the Bcd gradient (*Gregor et al., 2007b*; *Durrieu et al., 2018*). However, the full three-dimensional geometry of the embryo can be important (*Mogilner and Odde, 2011*), such as in interpreting FRAP (*Castle et al., 2011*) measurements and explaining differences in the gradient between dorsal and ventral sides of the embryo (*He et al., 2010a*; *Hengenius et al., 2011*). Here, we perform our modeling in three-dimensions as our experimental perturbation affects the three-dimension geometry of embryos.

Considering the Bcd concentration *[Bcd]*, Bcd diffusion (D), Bcd degradation rate (μ) and Bcd production rate (*J*), the SDD model is described by

$$\partial_t [Bcd] = D\nabla^2 [Bcd] - \mu[Bcd] + Jf(x)$$

We use Bcd dynamic parameters D = 4 $\mu m^2 s^{-1}$ and μ = 0.0005 $s^{-1}$ consistent with the most recent estimates for Bcd dynamics (*Durrieu et al., 2018*). Note these parameters are based on an one-dimensional SDD model, as such models are typically a reasonable approximation to the formation of the Bcd gradient (*Little et al., 2011*). *f(x)* is defined such that little Bcd production occurs greater than 50 μm from the anterior pole (Materials and methods). We solve for *[Bcd]* on the surface of an ellipsoid, as Bcd transport through the yolk appears to be limited (*Gregor et al., 2007b*). For each simulation, we keep all dynamic parameters fixed and only change the embryo geometry. In particular, we assume that the Bcd production rate is unchanged in *fat2RNAi* embryos. Since the embryo volume does not markedly change, this means that the total amount of Bcd is similar in the model for both wild-type and *fat2RNAi* embryos. We change the width to maintain the experimentally measured aspect ratios (*Figure 1C*) and account for eGFP folding time (~50 min) (*Liu et al., 2013*). See Materials and methods for detailed description.

The SDD model can fit the observed Bcd gradient profiles in different embryonic geometries without requiring any change to the Bcd dynamics or production (*Figure 1K*). As a consequence, we can explain the above observations. First, the lower Bcd intensity near the anterior pole in *fat2RNAi* embryos is because of dilution due to the larger DV extent in rounder embryos. If the total Bcd molecule number is similar at a particular AP position, the measured Bcd nuclear concentration (which depends on both total protein number and the local volume) at that position is lower in *fat2RNAi* embryos. Second, since the diffusion coefficient appears to remain unchanged, there is greater accumulation of Bcd in the posterior region of *fat2RNAi* embryos as the total travel distance is reduced in the rounder, shorter embryos, even when accounting for the higher curvature in *fat2RNAi* embryos. The ability of the SDD model to explain these differences in the Bcd profile simply by accounting for embryo geometry provides further support to the SDD model being an excellent biophysical description of the process of Bcd gradient formation. It is worth noting that the nuclei respond to the local concentration of mature Bcd protein regardless of the folding state of the tagged eGFP. Although we cannot infer the precise profile of mature Bcd from our experimentally measured eGFP profile as the Bcd protein folding rate is unknown, our model can test the effects of different protein folding time on the functional gradient in different geometries. From the model shown in *Figure 1K*, we can infer the total Bcd concentration, where we assume Bcd folds at a much faster rate than eGFP (*Figure 1—figure supplement 1G*). The predicted total Bcd concentration displays qualitatively similar behavior in varying embryonic geometries to our experiments; that is our results are unlikely an artefact of protein folding time differences between Bcd and eGFP.

Does the change of the Bcd concentration profile in *fat2RNAi* embryos impact on the scaling of downstream patterning genes? To address this question, we measured Hunchback (Hb) expression in mid n.c. 14 using live imaging of *hb >LlamaTag* (*Bothma et al., 2018*), (*Figure 2A*). In control embryos, the Hb expression boundary locates at 49.0% EL with variation of 1.3% EL, consistent with previous reports (*Houchmandzadeh et al., 2002*). Comparatively, the Hb boundary shows a posterization in *fat2RNAi* embryos (52.9% EL). Further, we found an increased variation of 2.3% EL in the Hb boundary position (*Figure 2B*). However, considering the absolute length of *fat2RNAi* embryos, the variation of Hb boundary positions (~8 μm) between different individuals is still less than the average distance between neighboring nuclei.

Using immunofluorescence, we next investigated the impact of geometrical perturbations on the expression domains of other gap genes. In agreement with our results for Hb, the boundary positions of all four gap genes displayed slight shifts in the posterior direction in *fat2RNAi* embryos (*Figure 2C–F*). However, the inter-individual variation remains comparable to that of the wild type (*Figure 2C–F*, bottom row). We conclude that when we manipulate the embryonic geometry to an extent beyond that naturally observed, the reproducibility of the patterning outcomes is preserved. Therefore, the intact early embryonic patterning network is highly robust to variations in embryonic geometry.

## Embryonic length predetermines patterning outcomes in decanalized conditions of increased bcd dosage

We have demonstrated that *fat2RNAi* embryos provide an excellent system for testing the role of geometry on patterning networks. Taking advantage of the availability of genetic manipulations in *Drosophila*, we now use this tool to explore the role of geometric constraints in determining phenotypic outcomes in decanalized conditions: first, in embryos with increased maternal *bcd* gene dosage; and second in embryos with depleted maternal *bcd*.

Phenotypic discordance has been observed previously as a consequence of artificially altered maternal *bcd* dosage. Gradual increase of the maternal *bcd* gene copies leads to a larger proportion of individuals showing defective patterning (*Namba et al., 1997*). We wanted to test our hypothesis that the embryonic geometry predetermines patterning outcomes of these individuals. To efficiently increase the Bcd gradient amplitude, we generated a tandem *bcd* construct, where two copies of the *bcd* gene are linked by the P2A self-cleaving peptide (*Figure 3A*). Two transgenic fly lines with two and four genomic insertions of this construct deposit *bcd* mRNA into embryos at ~3 (6x *bcd*) and ~5 (10x *bcd*) fold wild-type amounts, respectively (*Figure 3B*; *Figure 3—figure supplement 1A–C*). As Bcd protein counts scale linearly with that of its mRNA, the corresponding amplitude of the Bcd gradient are expected to show the same fold changes (*Petkova et al., 2014*), as manifested by the posterior displacement of cephalic furrow position (*Figure 3B*).

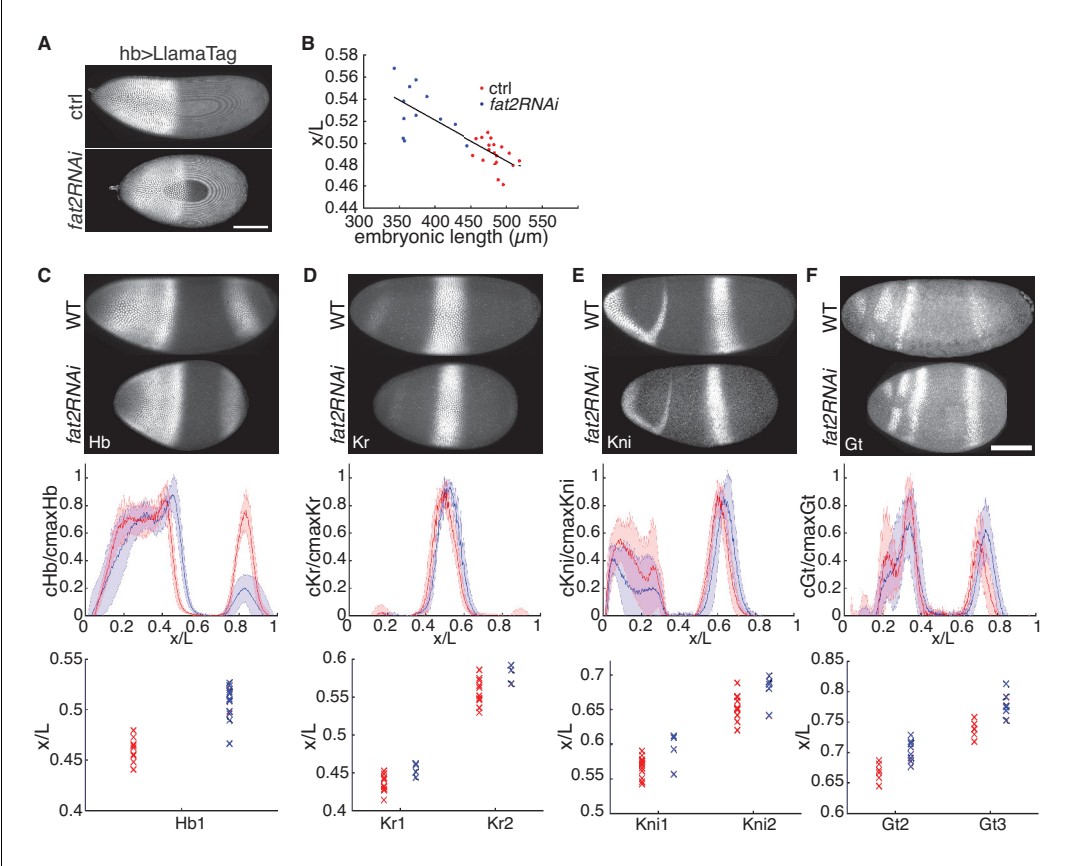

**Figure 2.** Embryonic patterning is robust to perturbation of embryonic geometry. (**A**) Max projection of ctrl (top) and *fat2RNAi* (bottom) embryos expressing maternally loaded eGFP and hb >LlamaTag. (**B**) Scaled Hb boundary positions plotted against EL in two genotypes. (**C–F**) Comparison of gap gene expression between wild type and *fat2RNAi* embryos. Profiles normalized to max intensity and the computed mean and s.d. plotted against scaled AP length. Boundary positions of each individual is plotted to show the distribution in two genotypes (WT, red and *fat2RNAi*, blue). Scale bars, 100 μm.

The ~5 fold *bcd* overexpression compromises viability to adulthood (*Figure 1—figure supplement 1D*) and the non-hatched embryos displayed a plethora of defective patterning phenotypes (*Figure 3C*; *Figure 3—figure supplement 1D*). Individuals with mild defects frequently displayed missing or fused denticle belts in the fourth abdominal (A4) segment (*Figure 3C–D*), a positional bias that has been reported previously (*Namba et al., 1997*). More severe phenotypes showed defects in a spreading region centered about the A4 segment. Meanwhile, embryos show high rate of mouth defects as a consequence of significantly increased local Bcd concentration in the most anterior region (*Figure 3C–D*; *Figure 3—figure supplement 1D*). Patterning defects were rarely seen in 3-fold *bcd* over-expression embryos unless *fat2RNAi* knockdown is further introduced into this genetic background (*Figure 3C–D*; *Figure 3—figure supplement 1E*). A large percentage of these individuals showed abdominal patterning defects, with A4 showing the highest defective frequency (*Figure 3C–D*). Interestingly, a similar distribution of defective abdominal segments is also seen in the small proportion of non-hatched *fat2RNAi* individuals (*Figure 3C–D*).

To understand if embryonic geometry predetermines the severity of phenotypic defects in individuals, we characterized the patterning outcomes using En expression in embryos with various *bcd* copy number and embryonic length. Individuals with 2-fold *bcd* overexpression (single insertion of tandem *bcd*), within the natural range of embryonic geometry, showed intact En expression. However, shorter embryos (<450 μm) frequently presented patterning defects, most commonly in the 6th En stripe (*Figure 3E*, top panel). Interestingly, this position corresponds to the A4 segment in the cuticle pattern. Comparatively, patterning defects become more pervasive in 3-fold *bcd* over-expression individuals when embryonic length reduced below 470 μm. The range of defective

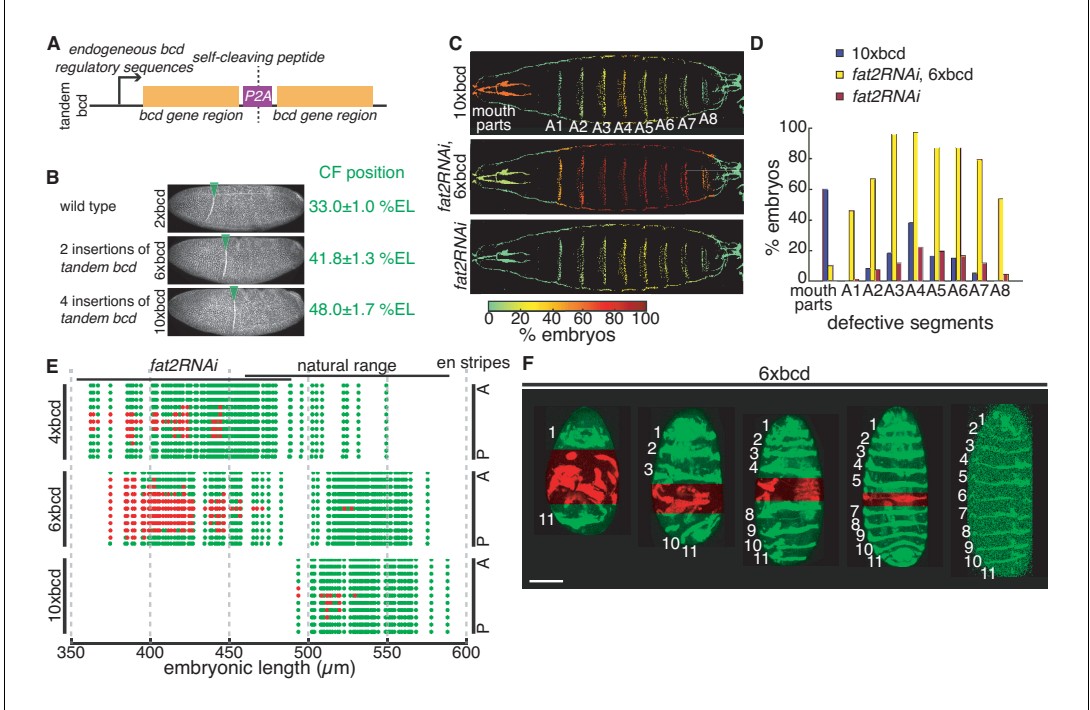

**Figure 3.** Defects due to decanalization by Bcd over-expression are length-dependent. (A) Schematic illustration of tandem *bcd* construct. (B) Embryos expressing 2x (wild type), 6x and 10x of maternal *bcd* fixed at the onset of gastrulation and stained with Phalloidin. Green arrowheads indicate the positions of cephalic furrow (CF) formation. (C) Distribution of defective cuticular segments in non-hatched 10x*bcd* (top), *fat2RNAi, 6xbcd* (mid) and *fat2RNAi* (bottom) embryos. Colormap indicates the frequency of defects in each segmental region. (D) Bar plots showing the distribution of defective cuticular segments in three genotypes. (E) En expression patterns in 4x (top), 6x (mid) and 10x (bottom) *bcd* embryos with different EL. Green dots, normal En stripes; red dots, defective En stripes; A, anterior; P, posterior. (F) Representative 6x*bcd* embryos with different EL expressing *en >mCD8:: GFP*. Numbers indicate En stripe identities and red mask indicate defective segmental regions. Scale bar, 100 µm.

The online version of this article includes the following figure supplement(s) for figure 3:

**Figure supplement 1.** Embryonic patterning with maternal *bcd* overexpression.

segments gradually expands from the 6th En stripe to both anterior and posterior regions with decreasing embryonic length (*Figure 3E*, middle panel and 3F). Further, increasing *bcd* dosage 5-fold renders patterning processes exceedingly susceptible to reduced embryonic length. Defects are observed in comparatively shorter individuals within the natural range and recurringly the 6th En stripe is the most frequent breaking point in the patterning (*Figure 3E*, bottom panel).

## The A4 segment is a weak point in the gap gene network due to repression of eve stripe five in short embryos

The defective abdominal patterning that we observe here is an intuitive result, as both the posterization of gap gene boundaries due to increased *bcd* dosage and reduced embryonic length lead to reduced number of nuclei along the AP axis in the trunk region. When the number of nuclei falls short of the minimal requirement to fulfill all the different cell identities along the AP axis, certain cell fates become lost. It is surprising, however, that this defect originates at and expands from the same position in all defective embryos, the A4 segment. This positional bias is also reflected in the segmentation gene pattern at the blastoderm stages. While the gap gene boundaries remain roughly at the same scaled positions across different geometries (*Figure 4A*), the absolute distance between neighboring gap gene expression peaks decreases in response to reduced embryonic length (*Figure 4B*). This in turn changes the combinatorial inputs to activate downstream pair-rule genes, for example *even-skipped* (*eve*). The expression peaks of Kni and Gt are brought into proximity with gradually reducing embryonic length (*Figure 4B–C*). As Kni and Gt confine the boundaries of *eve* stripe 5 (*Fujioka et al., 1999*), the expression of this *eve* stripe is over-repressed (*Figure 4C*, asterisk). This results in the loss of correct cell fate at this position, corresponding to the future A4

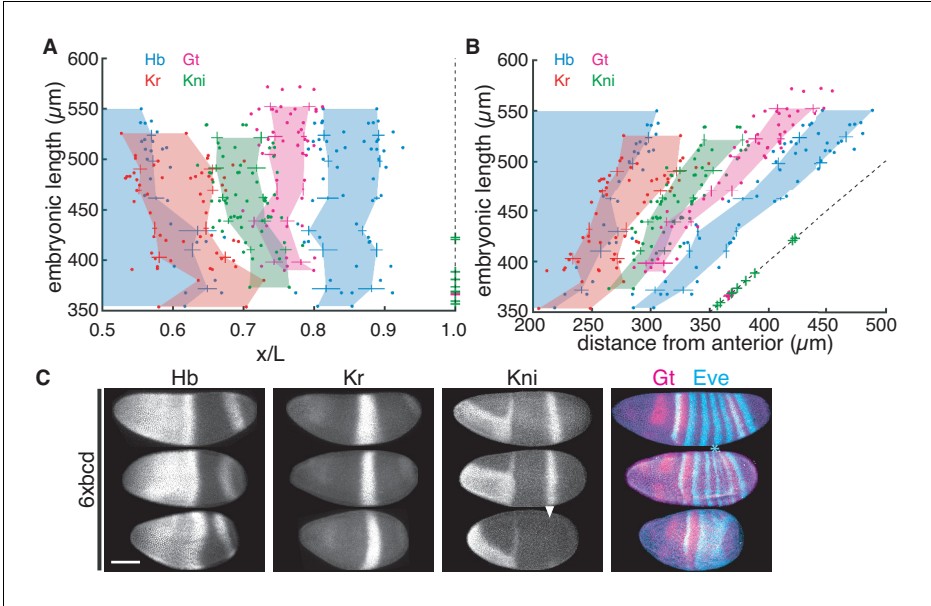

**Figure 4.** Embryonic patterning breaks down at A4 segment with *bcd* overexpression. (A–B) Embryonic length of *6xbcd* individuals plotted against the scaled (A) or absolute (B) AP position of four gap gene boundaries (Hb, blue; Gt, magenta; Kr, red and Kni, green). Data from every 30 μm EL interval were binned to compute mean and s.d. and the colored areas are generated by connecting mean values of different EL ranges. Dashed line indicates posterior boundary of individuals; green and magenta crosses overlapping the dashed line indicate individuals with corresponding EL not expressing Kni and Gt, respectively. (C) Representative segmentation gene expression in *6xbcd* embryos with different EL. Asterisk indicates repressed Eve stripe five and arrowhead indicates failed activation of Kni. Scale bar, 100 μm.

segment. With further reduced embryonic length, a larger percentage of individuals fail to activate Kni and Gt in the trunk region (*Figure 4B*, green crosses; *Figure 4C*, arrowhead), leading to defects across a broader range.

## AP patterning of bcd mutants correlates with embryonic length

We have shown that embryonic geometry predicts individual patterning outcomes under increased *bcd* dosage. To understand if embryonic geometry is a general factor underlying phenotypic discordance in decanalized conditions, we asked how geometrical perturbation influences phenotypic outputs in the absence of Bcd inputs. This question is motivated by the significant phenotypic variation observed among embryos derived from females carrying the same *bcd* null allele (*Frohnhöfer and Nüsslein-Volhard, 1986*; *Frohnhofer and Nusslein-Volhard, 1987*). For example, the individuals derived from *bcd*[E1] homozygous females show highly variable pattern in the perspective A1 to A5 segments, manifesting in either fusion or depletion of various number of denticle belts.

To systematically understand the inter-individual phenotypic variation among *bcd* mutants, we utilized an allele generated by the CRISPR-MiMIC method (*Huang et al., 2017*; *Venken et al., 2011*). The MiMIC transposon carrying stop codons in all three reading frames is targeted by CRISPR to insert into the first intron of the endogenous *bcd* gene. Therefore, no functional Bcd protein is produced by this knockout allele (annotated *bcd*[KO]). Moreover, the MiMIC construct contains an eGFP marker, which facilitates further genetic manipulation, such as recombination, carried out in this study.

The cuticular pattern of the *bcd*[KO] allele qualitatively recapitulates that of *bcd*[E1] (*Figure 5—figure supplement 1A–C*). While the anterior embryonic patterning is entirely defective, the patterning defects in posterior regions are more variable. The number of normal abdominal denticle belts in each embryo ranges from three (A6 - A8) to seven (A2 - A8), with four intact abdominal segments (A5 –A8) being the most frequently observed phenotype (*Figure 5A*). Structures indicating partially

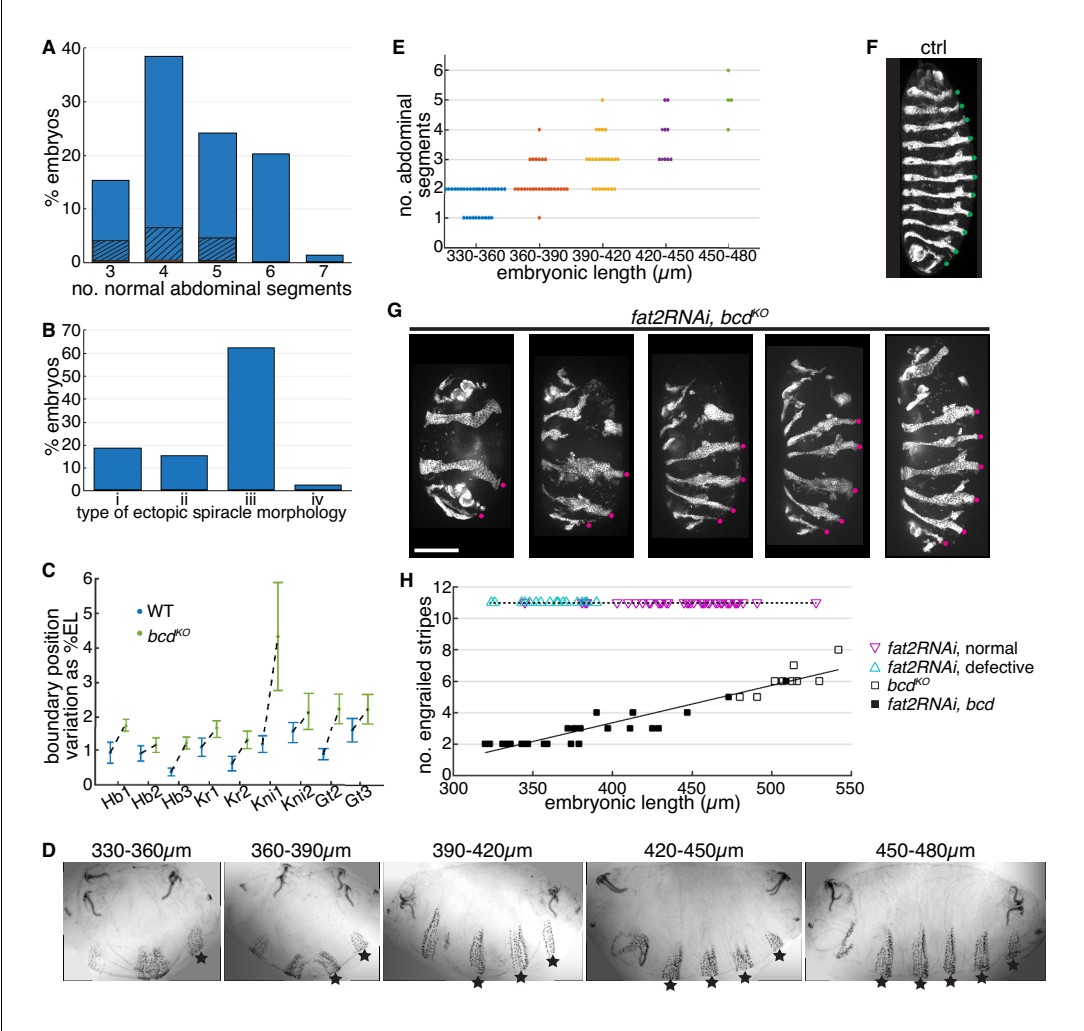

**Figure 5.** *bcd* mutant phenotypes correlate with embryonic length. (A) Phenotypic frequency showing different number of normal abdominal segments in *bcd^KO* mutant individuals (n = 202). Dashed area indicates proportion of embryos showing fully developed ectopic spiracles (see *Figure 5—figure supplement 1D(i)*). (B) Phenotypic frequency of different ectopic spiracle morphology as shown in *Figure 5—figure supplement 1D* (n = 168). (C) Variation of gap gene boundary positions in wild type vs. *bcd^KO* embryos. Error bars are computed by bootstrapping with data shown in *Figure 5—figure supplement 1E–H*. (D) Representative cuticle patterns of *fat2RNAi, bcd^KO* embryos within different ranges of embryonic length, from left to right, 330–360, 360–390, 390–420, 420–450, and 450–480 μm. Stars indicate normal abdominal segments. (E) Number of normal abdominal segments plotted against EL range of each individual. (F) En expression in wild type embryo. Green dots indicate En stripes. (G) Representative images of *fat2RNAi, bcd^KO* embryos showing different number of En stripes. Magenta dots indicate En stripes. Scale bar, 100 μm. (H) Number of En stripes plotted vs. EL in individuals from three genotypes. Magenta triangles indicate individuals showing defective morphogenesis at the end of dorsal closure; cyan triangles indicate normal morphogenesis.

The online version of this article includes the following figure supplement(s) for figure 5:

**Figure supplement 1.** Characterization of *bcd* mutant phenotypes.

**Figure supplement 2.** Variations of *bcd* mutant phenotypes.

**Figure supplement 3.** *Bcd* mutant phenotypes correlate with embryo length.

differentiated abdominal segments can be observed in the anterior regions of the *bcd^KO* embryos. However, these structures do not recapitulate any of the wild-type denticle belts (*Figure 5—figure supplement 1B*). Further, we observed and classified the variable phenotypes of the duplicated posterior spiracles according to the completeness of the organ morphogenesis (*Figure 5B*; *Figure 5—figure supplement 1D*). Interestingly, the fully developed ectopic spiracle organ (*Figure 5—figure supplement 1D* (i)) is only observed in embryos showing less than six intact abdominal segments, while most frequently observed in individuals with three intact abdominal segments (*Figure 5A*,

dashed bar area). This indicates correlation between the patterning outcomes of the abdominal regions and the ectopic spiracles.

The wide spectrum of $bcd^{KO}$ phenotypes can be attributed to variations in patterning gene expression during the blastoderm stage. By the end of the blastoderm stage, the relative boundary positions of all gap gene expression domains show significant anterior shift compared to those in the wild-type embryos (*Figure 5—figure supplement 1E–H*). Importantly, the absence of Bcd activity results in significantly increased variation in gap gene boundary positions (*Figure 5C*, p<0.01 for all measured boundaries to have increased error randomly). Occasionally we detected no Krüppel (Kr) nuclear intensity in the presumed expression region in $bcd^{KO}$ individuals (2 out of 15 individuals), indicating the failure of Kr gene activation in these embryos (*Figure 5—figure supplement 2A*). The anterior Gt domain shows similar inter-individual variation, with the majority of the embryos failing to properly activate anterior Gt expression (8 out of 10 individuals). Instead only a thin stripe of diminished cytoplasmic signal can be detected (*Figure 5—figure supplement 1H*, asterisk; *Figure 5—figure supplement 2B*). It is noteworthy that embryos derived from a single pair of $bcd^{KO}$ parents raised in constant environmental conditions show equivalent phenotypic variation (*Figure 5—figure supplement 2C–D*), suggesting that the inter-individual variation observed cannot be attributed to differences in either environment or genetic background.

Next, we introduced the *fat2RNAi* knockdown into a $bcd^{KO}$ background to see how embryonic patterning is affected. The cuticle pattern of embryos derived from *fat2RNAi, $bcd^{KO}$* females resembles that of $bcd^{KO}$ alone, but the number of properly patterned abdominal denticle belts reduces with decreasing embryonic length. Moreover, novel phenotypes showing only one or two abdominal segments were observed when the embryonic length drops beyond the natural range (*Figure 5D and E*). All of the *fat2RNAi, $bcd^{KO}$* embryos showed duplicated spiracles with fully developed morphology (see *Figure 5—figure supplement 1D* (i)), consistent with our previous result that such spiracles prevail in embryos with shorter AP length. We observed similar behavior in the pattern of En expression. In contrast to invariable eleven En stripes in control embryos (*Figure 5F*), the number of En stripes shows positive linear correlation with embryonic length in $bcd^{KO}$ embryos, both in individuals with natural embryonic geometry (*Figure 5—figure supplement 3*; *Video 3*) and *fat2RNAi* knockdown (*Figure 5G and H*).

## Embryonic length dictates gap gene expression patterns in the absence of bcd

What underlies the correlation between embryonic length and phenotypes of $bcd^{KO}$ embryos? We next focus on variation in gap gene expression patterns in $bcd^{KO}$ embryos of varying geometry. *Figure 6A* shows representative expression patterns of four gap genes in $bcd^{KO}$ and $bcd^{KO}$, *fat2RNAi* mutants. The gene network shows qualitative differences in behavior within different ranges of embryonic length. Without the long-range gradient of Bcd, zygotic *hb* transcription is activated by the termini system mediated by the terminal gap gene, *tailless* (*Margolis et al., 1995*; *Figure 6—figure supplement 1A*). As a result, two Hb stripes form near the anterior and posterior poles of the embryo, spanning a width of ~10 and 15 nuclei, respectively. In embryos with extremely large aspect ratio (range 1, EL within 330–360 μm), the two Hb expression domains are in close proximity. This inhibits the expression of Kni, which is strongly repressed by Hb, in the central region of the embryo (*Hülskamp et al., 1990*). Meanwhile, Gt is activated by uniformly distributed maternal Cad protein, and in turn inhibits Kr expression (*Kraut and Levine, 1991*; *Rivera-Pomar et al., 1995*; *Figure 6A–C*, range 1).

In individuals with moderately increased embryonic length (range 2, EL within 390–420 μm), Hb stripes in the terminal regions separate further apart, permitting Kni expression in the middle region (*Figure 6A–C*, range 2). This Kni

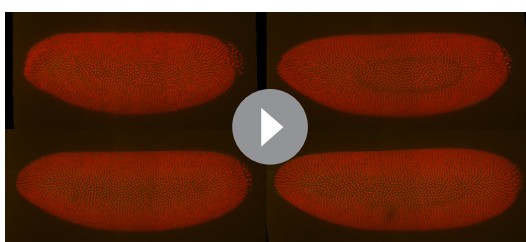

**Video 3.** Engrailed expression in $bcd^{KO}$ mutant individuals. Live imaging of $bcd^{KO}$ embryogenesis from onset of gastrulation to the end of dorsal closure. Embryos express H2Av::mCh (red) and en>mCD8::GFP (green). Dots indicate En stripes and the numbers on top of the embryos indicate the AP length of each individual.
https://elifesciences.org/articles/47380#video3

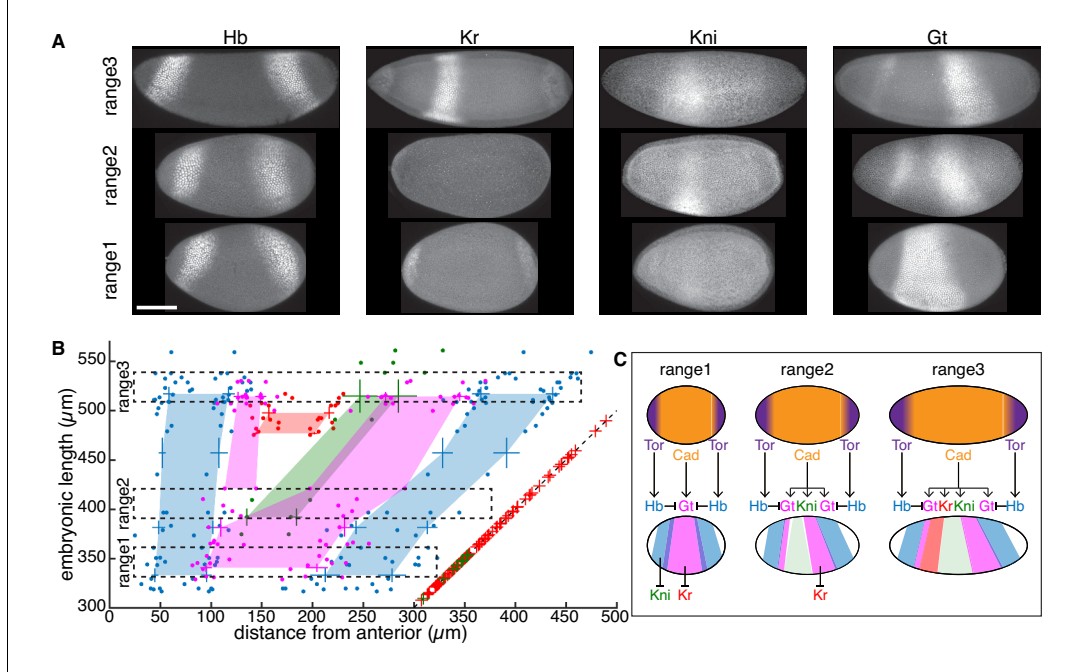

**Figure 6.** Phenotypic discordance can be traced back to variations in gap gene expression in $bcd^{KO}$ individuals. (**A**) Representative gap gene expression patterns within different range of EL. Range1, 330–360 µm; range2, 390–420 µm; and range3, 510–540 µm. (**B**) Embryonic length of $bcd^{KO}$ individual plotted vs. boundary position of four gap genes shown as absolute distance from the anterior pole (Hb, blue; Gt, magenta; Kr, red and Kni, green). Data from every 30 µm EL interval were binned to compute mean and s.d. and the colored areas are generated by connecting mean values of different EL ranges. Dashed boxes indicate ranges of EL corresponding to (**A**). Dashed line indicates posterior boundary of individual embryos; red and green crosses overlapping the dashed line indicate individuals with corresponding EL not expressing Kr and Kni, respectively. (**C**) Schematic illustration of positional information transfer from maternal systems to gap gene expression in $bcd^{KO}$ embryos within different range of embryonic length. Scale bar, 100 µm.

The online version of this article includes the following figure supplement(s) for figure 6:

**Figure supplement 1.** Positional information transfer in the absence of maternal *bcd*.

**Figure supplement 2.** Strong correlation between gap gene boundary positions and embryonic length in the absence of maternal *bcd*.

stripe is sandwiched by two Gt expression domains, a thin anterior stripe and a wider posterior one. The anterior Gt stripe in *bcd* mutants has been observed before (*Staller et al., 2015*) but its regulatory interactions remain elusive. Potentially it is activated by the remnant anterior determinants such as the maternal Hb, distributed in the anterior half of the embryo (*Irish et al., 1989*). Comparatively shorter embryos show phenotypically higher degree of symmetry in both cuticle and gene expression patterns, conceivably due to stronger repression of maternal Hb in shorter individuals (*Figure 6—figure supplement 1B–D*; *Figure 6—figure supplement 2*).

Looking more closely at individuals within the natural range of embryonic geometry (range 3, EL within 510–540 µm), the sufficient physical space between two Hb stripes permits the expression of Kr, Kni and Gt, arranged in spatial order that is conserved as in wild-type embryos (*Figure 6A–C*, range 3). In summary, as a consequence of gradually increasing embryonic length, a continuously increasing variety of gap gene expression domains are activated along the AP axis, which is in turn, translated into increased number of body segments, as manifested by the pair-rule gene expression pattern (*Figure 6—figure supplement 1E*).

## Discussion

Individuals of the same species often display a certain level of morphological and behavioral differences, such as in animal color patterns and human facial features (*Beldade et al., 2002*; *Sheehan and Nachman, 2014*). This reflects inter-individual variation in genetic composition and life-history environmental exposure (*Dall et al., 2012*). Such intraspecific individuality may have

significant ecological and social impacts on the population (*Forsman and Wennersten, 2016*). Equally, these same genetic and environmental variations pose challenges to fundamental developmental processes as they try to generate invariant developmental outcomes. Multiple lines of evidence suggest that organisms have evolved canalization mechanisms that render developmental processes insensitive to such sources of variation (*Félix and Barkoulas, 2015*; *Flatt, 2005*). Early *Drosophila* embryonic patterning provides an excellent example of a canalized developmental process – the boundaries of segmentation gene expression remain highly reproducible amongst individuals in the face of heterozygous mutations (*Lehmann and Nüsslein-Volhard, 1987*; *Wieschaus et al., 1984*), genetic variations (*Lott et al., 2007*) and temperature perturbations (*Houchmandzadeh et al., 2002*; *Lucchetta et al., 2005*). These studies suggest that mechanisms including epistasis, genotype-environment interactions and canalizing gene regulatory networks (*Manu et al., 2009*) work together to ensure precise patterning outputs.

In this study, we have identified embryonic geometry as an additional source of variation that patterning processes have evolved to buffer against. The geometry, or in other words, the aspect ratio of each ellipsoid-shaped embryo is determined during oogenesis, and this parameter varies by ±10% in the population of the wild-type strain OreR. The variable geometry in turn increases the variation in embryonic length given the natural range of embryonic geometry. Previous studies have shown that patterning outcomes are highly reproducible and remain scaled to embryonic length (*Antonetti et al., 2018*; *Lott et al., 2007*). What roles do Bcd and the gap gene interactions have in this scaling (*Wu et al., 2015*)? Combining results from *fat2RNAi* embryos with theoretical modeling may reveal exciting new insights into the underlying scaling mechanisms (*Bieler et al., 2011*; *Jaeger et al., 2004*). Correspondingly, we found that under decanalized conditions, either by depleting maternal *bcd* inputs or artificially increasing the *bcd* dosage, the patterning process loses its capacity to buffer embryonic length variations. Consequently, the length of an individual embryo predetermines its patterning outcomes. The predictive power becomes stronger when we artificially increase the variation of the embryonic geometry. The aspect ratio of the *fat2RNAi* embryos differs by ±30% while the average embryonic volume is only slightly reduced by ~8%. These results further support embryonic geometry as a major source of variation that accounts for inter-individual phenotypic variation under decanalized conditions.

Both embryonic volume and embryonic length are inheritable traits and therefore adaptive to artificial selection or environmental changes (*Azevedo et al., 1997*; *Fox and Czesak, 2000*; *Lott et al., 2007*; *Miles et al., 2011*). It will be interesting to understand if the aspect ratio of the embryo shape is also a genetically variable trait so that the population can be selected to produce progenies with a biased geometry. If this is the case, embryonic geometry may be involved in the complex interplay between environment, genetic components and developmental processes during the course of evolution. When a population confronts selection towards a new phenotypic optimum, for example, larger egg volume due to decreasing temperature (*Azevedo et al., 1997*), such directional selection may result in decanalizing effects on the patterning processes (*Miles et al., 2011*; *Wagner et al., 1997*). Meanwhile, the naturally variable embryonic geometry - together with other sources of variation - generates a spectrum of patterning outcomes in different individuals. As a result, a different range of embryonic geometry will be favored and selected as they maintain the patterning outcomes of the parental lines. Conceivably, this may be one of the reasons why eggs of closely related Dipteran species differ not only in size (*Gregor et al., 2008*; *Markow et al., 2009*) but also in geometry, and such geometrical differences can also be observed in different laboratory lines carrying different genetic background.

Our quantitative analysis of segmentation gene expression demonstrates how embryonic geometry affects individual patterning outcomes under two decanalizing conditions. In the case of the maternal *bcd* null mutant, we have shown that the signaling centers located at both poles of the embryo initiate the hierarchical gene expression along the AP axis in a non-scaled manner. This explains, mechanistically, how patterning processes incorporate information of the embryonic geometry to account for the final outputs. It remains unclear, however, in the case of increased *bcd* dosage, what determines the breaking point (the fourth abdominal segment) of the final pattern. One possibility is that the susceptibility of this position reflects the strength of the regulatory interactions between the segmentation genes (*Jaeger, 2011*). Systematic comparisons among different *Drosophila* species have shown that the regulatory sequences of the segmentation genes are rapidly evolving and thus substantially diverged (*Ludwig et al., 2000*). Interestingly, the spatio-temporal

dynamics of the segmentation gene expression patterns are highly conserved between species, suggesting that the co-evolution of modular transcription binding sites compensate for each other to keep the patterning outcomes unchanged (*Gregor et al., 2005*; *Ludwig et al., 2005*; *Ludwig et al., 2000*). Such an inter-species canalization phenomenon is also observed among more distally related species within the sub-taxon Cyclorrhapha, which involved more dramatic rewiring of the regulatory network (*Crombach et al., 2016*; *Wotton et al., 2015*). If the breaking point of patterning processes under decanalized conditions truly depends on the system parameters of the underlying network (*Jaeger and Crombach, 2012*), we expect to see different susceptible points in different network structures. This can be tested by characterizing decanalizing phenotypes in related species.

A longstanding question in patterning is how do gene regulatory networks downstream of morphogens incorporate information about the macroscopic geometrical parameters of each individual to give rise to scaled patterning outputs? While our results do not provide mechanistic insight into the scaling mechanisms of the gap gene network, they define the physical boundaries where such scaling breaks down. The emergence of scaled expression boundaries is closely linked to embryo geometry, and future models will hopefully more rigorously test the role of geometry in the scaling. Further, our results directly show that the Bcd gradient shape depends on the embryo geometry; in our case, by roughly conserving embryo volume but reducing the aspect ratio, the Bcd gradient extends further into the posterior, resulting in a posterior shift in gap gene expression boundaries.

In conclusion, embryonic geometry was identified as a source of variation in addition to environmental and genetic factors that predetermines phenotypic outcomes in mutant conditions. We think that embryonic or more generally, tissue geometry may play an important role in other decanalizing conditions by affecting patterning outputs, such as other segmentation gene mutants (*Janssens et al., 2013*; *Surkova et al., 2013*), or in vitro induction of patterning systems (*Lancaster and Knoblich, 2014*; *Simunovic and Brivanlou, 2017*), both of which show significant inter-individual phenotypic variations. Our work highlights that care must be taken when taking a system out of its native environment – for example organoids – as the system boundaries affect the operation of signaling networks. Characterizing the influence of the geometrical parameters will help us to have a more complete understanding of decanalization, and in turn, canalizing phenomenon.

# Materials and methods

## Fly stocks and genetics

The *bcd* knockout allele (*bcd*$^{KO}$) used in this study was generated by CRISPR-mediated insertion of a MiMIC cassette into the first intron of the *bcd* gene (*Huang et al., 2017*; *Venken et al., 2011*) The cuticle phenotype of *bcd*$^{KO}$ was compared to that of the classic *bcd*$^{E1}$ allele (*Frohnhöfer and Nüsslein-Volhard, 1986*) To generate embryos with artificially reduced aspect ratio, we expressed RNA interference against the *fat2* gene using a maternal *traffic jam (tj) >Gal4* driver (*Barlan et al., 2017*) Both *UAS > fat2* RNAi and *tj >Gal4* were either crossed to or recombined with the *bcd*$^{KO}$ allele, so that the females carrying all three alleles produce bcd null embryos with wide range of aspect ratio.

The *tandem bcd* construct was generated by replacing the eGFP sequence in the pCaSpeR4-*egfp-bcd* vector (*Gregor et al., 2007b*) by *bcd* protein coding sequence. First, the vector was digested with NheI and SphI to remove the eGFP. Next the *bcd* coding sequence was amplified by PCR from the vector using primer pairs

5'-cggagtgtttgggctagcaaagatggcgcaaccgccg-3' and
5'-gttagtagctccgcttccattgaagcagtaggcaaactgcgagtt-3'.

Further the P2A self-cleaving peptide with the GSG linker was synthesized as oligo pairs

5'-tttgcctactgcttcaatggaagcggagctactaacttcagcctgctgaagcaggctggagacgtggaggagaaccct
ggacctgcatgcatggcgcaaccgc-3' and
5'ggcggttgcgccagcatgcaggtccagggttctcctccacgtctccagcctgcttcagcaggctgaagttagtagctccgcttc-
cattgaagcagtaggcaaa-3'.

These two fragments and the digested vector were then assembled using Gibson Assembly strategy (NEB). The final construct was injected (BestGene Inc) and two insertions on 2nd (*tdBcd(II)*) and 3rd (*tdBcd(III)*) chromosome, respectively, were established and used for this study. Consequently,

hetero- or homo-zygous *tdBcd(III)* females produce embryos with 4x and 6x of maternally loaded Bcd protein, respectively (compared to 2x*bcd* in the wild-type); and homozygous *tdBcd(II);tdBcd(III)* females generate 10x*bcd* embryos. Finally, females homozygous for *tdBcd(III)* which also carry *tj >Gal4* and *UAS >fat2* RNAi generate 6x*bcd* embryos with reduced aspect ratio.

Other fly lines used in this study include a laboratory OreR strain raised in 25℃ (the wild-type control); *en >mCD8* GFP (to visualize dynamic *en* expression pattern); *egfp-bcd* line (for quantification of Bcd gradient profile); *mat >eGFP*; *hb >LlamaTag* (gift from Hernan Garcia's lab); *Df(3R)tll^g* (BL#2599).

## Measurement of embryonic geometry

To compare the geometrical parameters between OreR, *bcd^KO* and *fat2RNAi* populations, embryos were dechorionated and aligned laterally on an agar plate and imaged under a stereoscope (Nikon SMZ18). Images were then segmented to extract the embryo contour and fitted to elliptic shapes. The long and short axes of fitted ellipses were taken as the measurement of embryonic length and width, respectively. The approximate embryonic volume is calculated using the measured length and width, assuming embryos are ellipsoids in shape. For each strain, more than 200 individuals were measured.

For confocal live imaging, embryonic length was measured as the longest distance between the anterior and posterior poles. The immunostaining procedures result in an isotropic shrinkage of embryonic volume. To measure the geometrical parameters of the fixed embryos, we first carried out a linear fit between aspect ratio and embryonic length using stereoscope data. Further we measured the aspect ratio of each fixed embryo and estimate its embryonic length and width using the same linear fit equation.

We checked our assumption of isotropic shrinkage upon fixation (*Figure 1C* and *Figure 1—figure supplement 1A*). Comparing embryos from live imaging and fixed embryos, we did not observe significant anisotropy in the embryo geometry (*Figure 1—figure supplement 1B–C*).

## Immunostaining

Embryos at desired stages were dechorionated by household bleach and fixed in heptane saturated by 37% paraformaldehyde (PFA) for 1 hr. The vitelline membrane was subsequently manually removed. Prior to incubation with primary antibodies, embryos were blocked with 10% BSA in PBS. Antibodies used were guinea pig anti-Hb (1:2000), rabbit anti-Gt (1:800), guinea pig anti-Kr (1:800), guinea pig anti-Kni (1:800), guinea pig anti-Eve (1:800). Primary antibodies were detected with Alexa Fluor-labelled secondary antibodies (1:500; LifeTech). Embryos were co-stained with Phalloidin conjugated with Alexa Fluor for staging purpose or visualizing cephalic furrow position. Short incubation of Dapi dye was carried out during the last wash prior to mounting to visualize presyncytial nuclei. Embryos were mounted in AquaMount (PolySciences, Inc) and imaged on a Zeiss LSM710 microscope with a C-Apochromat 40x/1.2 NA water-immersion objective. Hb, Gt, Kr, Kni and Eve antibodies were gifts from Johannes Jaeger.

## Cuticle preparation

Embryos of various genotypes were collected during the blastoderm stages and allowed to develop at 25℃ until the end of embryogenesis. The embryos were then dechorionated, fixed, devitellinized and incubated into a mixture of Hoyer's medium and Lactic acid in a 1:1 ratio at 65℃ between an imaging slide and a cover slip. For an exhaustive description of the method used see *Alexandre (2008)*.

## Measurement of bcd profile

For measurement of Bcd gradient profile, we followed the protocols detailed in *Gregor et al. (2007a)*. Embryos expressing eGFP-Bcd either with or without *fat2RNAi* were dechorionated and mounted laterally on a confocal microscope (Zeiss LSM710). The images were acquired at the mid-sagittal plane of embryos at early n.c. 14. Data acquired in *Figure 1F–H* for different individuals (including eGFP:Bcd and eGFP:Bcd; *tj >Gal4,UAS > fat2* RNAi) were co-mounted on the same glass-bottom dish and taken with identical microscope settings. For each image, nuclear centers along the dorsal edge of the embryo were manually selected and the corresponding circular area was used to

compute the average fluorescent intensity. Nuclear intensity was then plotted against either absolute distance from the anterior or scaled AP position. To compare average profiles between control and *fat2RNAi* embryos, all nuclei from embryos either longer or shorter than 450 µm are binned in 50 bins along the scaled AP axis over which the mean and standard deviation were computed.

## Fluorescence in situ hybridization

### Probe synthesis

A 471 bp region of *bcd* transcript was amplified from early embryo cDNA using primer pairs:
5'-cccggatccCTCAAATAGCAGAGCTGGAGC-3' and 5'-cccggtaccGCTGCTGCTGGAAGAACTG-3' and subcloned into the pSP18 vector. Further, the vector was linearized by BamHI restriction digestion and DIG-labeled 'anti-sense' RNA was synthesized with T7 RNA polymerase using DIG RNA Labeling Kit [Roche-11175025910]. Synthesized RNA probes were then precipitated by adding 75 µl pre-chilled ethanol, 1.3 µl LiCl (7.5 M) and 1 µl yeast tRNA (25 mg/ml) to the labeling reaction. Precipitation takes overnight at −20˚C. After centrifuging at 4˚C full speed for 30 min, the precipitated RNA probes were washed with 70% ethanol (in DEPC water) and re-suspended in 100 µl Hyb-A buffer (50% formamide, 5XSSC buffer, 100 µg/ml salmon DNA [BDL F012], 50 µg/ml heparin [Sigma-Aldrich H4784], 0.1% tween-20 in DEPC water). The synthesized RNA probes were stored in −20˚C.

### Pre-hybridization

Embryos at stages within 2 hr after fertilization were collected, dechorionated and fixed in 1:1 solution of heptane and 4% formaldehyde in PBS for 20 min. After that, lower phase of the solution was replaced by methanol with the same volume. The solution was vortexed for 20 s to remove the vitelline membrane, and the embryos at the bottom of the lower phase were collected and rinsed three times with methanol. Embryos were then rehydrated by washing them in sequentially increasing percentage of PBT (PBS+0.1%Tween-20) in methanol. Next, embryos were post-fixed in 4% formaldehyde in PBT for 20 min and washed 5 times in PBT. Embryos were then washed in Hyb-B buffer (50% formamide, 5XSSC in DEPC water). Finally, pre-hybridization was carried out by placing the embryos in Hyb-A buffer at 65˚C for 3.5 hr.

### Hybridization

5 µl of RNA probes was diluted in 250 µl Hyb-A buffer. The probe mix was heated at 80˚C for 10 min and placed on ice for 5 min. The probe mix was added to the embryos and hybridization was carried out at 65˚C for 18 hr. The embryos were washed at 65˚C in Hyb-B buffer for 6 times, 30 min each wash. After, the embryos were washed sequentially in the following solution at room temperature: 4:1 Hyb-B:PBT, 1:1 Hyb-B:PBT, 1:4 Hyb-B:PBT and PBT, 5 min each wash. Embryos were incubated with blocking solution (4:1 PBT:Western blocking reagent (Roche- 11921673001)) for 1 hr. For primary staining against DIG-labeled RNA probes, embryos were incubated with sheep anti-DIG [Sigma-Aldrich 11333089001] with 1:400 dilution in the blocking solution at 4˚C overnight. After washing in PBT for 6 times with 20 min each wash, embryos were stained with the secondary antibody donkey anti-sheep Alexa 555 [Thermo Fisher Scientific A21436] with 1:500 dilution in PBT for 1 hr at room temperature. Finally, after washing in PBT for 6 times, embryos were mounted on microscopy slides with Aqua-Poly/Mount [Polysciences, Inc- 18606].

### Microscopy imaging and image analysis

Slides were imaged on a Zeiss confocal LSM880 using 40X water immersion lens. For each embryo, Z-stack images were taken covering the region from surface to the midsagittal plane, and the neighbouring planes were separated by 3 µm. Following that, a projected maximum intensity image was generated. The outside of the embryo was segmented using thresholding in Matlab after using rolling ball background subtraction. A 10µm-thick mask was defined around the embryo perimeter in order to avoid effects from yolk autofluorescence. Finally, the average signal intensity at each position along the AP axis was taken from the mask. The signal from each embryo varies significantly, making a rigorous quantitative comparison challenging. The signal from each embryo was normalized by the intensity around 200 µm from the anterior pole.

## Simulation of SDD (Synthesis, Diffusion, Degradation) model

We use COMSOL Multiphysics 5.3 to simulate diffusion along the surface of prolate ellipsoids of varying sizes to account for the geometries of wild type and *fat2RNAi* embryos.

We consider steady state equations for the concentration of Bcd in the unfolded (U) and folded (F) states. These take the form:

$$\partial_t[U] = D\nabla^2[U] - (\mu + \alpha)[U] + Jf(x).$$

$$\partial_t[F] = D\nabla^2[F] + \alpha[U] - \mu[F],$$

where: $\nabla^2$ represents the Laplacian on the ellipsoid surface; $\alpha$ is the GFP folding rate; $\mu$ is the decay rate of Bcd in both the unfolded and folded forms; and $Jf(x)$ corresponds to a production rate of unfolded Bicoid, which we suppose to depend only on the distance to anterior (*x*-axis corresponds to AP direction). We consider a step-like production function:

$$f(x) = 1 - \tanh\left(\frac{(x-\lambda)}{\delta}\right),$$

where $\lambda$ is the AP-axis extension of a region of approximatively constant production; $\delta$ is a smoothing parameter. We have checked that the simulations weakly depend on the value of $\delta$ as long as $\delta << \lambda$.

Physical parameters were fixed based on previous literature, see Table below. Quantitative fits were obtained by using a L2-measure with a window-type weight function for data-points within the rescaled length range $x/L \in [0.2, 0.8]$.

The solution for folded Bcd is not flat is the vicinity of $z = 0$ (*i.e.* near the head); this is due to the projection into the Cartesian coordinates, leading to a singular integration measure near its extremal value $z = 0$. Similarly, we disregard the experimentally measured values of Bcd intensity within the first microns form the anterior pole as these are prone to large errors.

| Parameter | | Values | Reference |
|---|---|---|---|
| $D$ | Diffusion coefficient of folded and unfolded bcd | 4 $\mu m^2 s^{-1}$ | *Durrieu et al., 2018* |
| $\lambda$ | Production region | 50 $\mu m$ | in situ data in this study and *Little et al., 2011* |
| $\delta$ | Width of decay of the production region | 5 $\mu m$ | in situ data in this study |
| $\mu$ | Protein decay rate | 1/ (35*60) s | *Durrieu et al., 2018* |
| $\alpha$ | Folding rate | 1/ (50*60) s | *Durrieu et al., 2018* and *Liu et al., 2013* |
| $a_{WT}$ | WT type length, dorso-ventral axis | 100 $\mu m$ | See *Figure 1* |
| $c_{WT}$ | WT type length, anterio-posterior axis | 250 $\mu m$ | See *Figure 1* |
| $a_{fat2}$ | *fat2RNAi* type length, dorso-ventral axis | 115 $\mu m$ | See *Figure 1* |
| $c_{fat2}$ | *fat2RNAi* type length, anterio-posterior axis | 175 $\mu m$ | See *Figure 1* |

As the equations are solved on a closed surface, there are no explicit boundary conditions.

## Gap gene boundary quantification

Confocal Z-stack images were Z-projected (maximum intensity) in Fiji (RRID:SCR_002285) for further analysis. The images of laterally oriented embryos were rotated so that the anterior is to the left and dorsal to the up. A line with the width of 100 pixels crossing the center of the embryo was drawn to extract average intensity along the AP axis. The intensity profile was shown as a function of percent embryonic length (%EL). We determined the background intensity $I_{min}$ and maximum intensity $I_{max}$ of each embryo; $I_{min}$ was subtracted from the intensity profile and the resulting curve was rescaled by $1/(I_{max}-I_{min})$. The boundary position is defined where rescaled intensity profile equals to 0.5.

To estimate the variation of boundary position, we performed bootstrapping using the Matlab function *bootstrp*. We performed 1000 simulation runs to infer the variability on the boundary precision. In each run, a random data set was generated by sampling from the known experimental data. In this procedure, each data point can be included multiple times within each random sample. As

the number of samples per boundary was small, we did not test for significant changes in the precision of boundary specification for a single boundary between wild-type and $bcd^{KO}$ embryos. However, pooling the data from the different boundaries, we observe that precision in all the measured boundaries decreases (*i.e.* the error increases) in $bcd^{KO}$ embryos. We calculated the p-value using the paired sample t-test across all boundaries.

### Quantification of maternal bcd transcripts

To compare the relative amount of maternally loaded *bcd* transcripts in different genotypes, we extracted total mRNA from presyncytial embryos (within 1 hr after egg deposition) generated by OreR, *fat2RNAi*, *tdBcd(III)* or *tdBcd(II);tdBcd(III)* females and reverse transcribed to cDNA. We performed qRT-PCR with *bcd*-specific primer pair using SYBR Green (Thermo Fisher) protocol and the housekeeping gene *rpl32* was used as internal reference. The relative *bcd* mRNA amount was normalized to that of OreR. Three independent measurements were carried out over which the mean and standard deviation was calculated.

## Acknowledgements

We thank Sally Horne-Badovinac and Hernan Garcia for fly lines. We thank Alexis Kerh for help with fly work. We thank Jean-Paul Vincent for support in revising the manuscript. This work was supported by a National Research Foundation Singapore Fellowship awarded to TES (NRF2012NRF-NRFF001-094) and funding from the Mechanobiology Institute, National University of Singapore, Singapore.

## Additional information

### Funding

| Funder | Grant reference number | Author |
| --- | --- | --- |
| National Research Foundation Singapore | NRF2012NRF-NRFF001-094 | Timothy E Saunders |

The funders had no role in study design, data collection and interpretation, or the decision to submit the work for publication.

### Author contributions

Anqi Huang, Conceptualization, Formal analysis, Validation, Investigation, Visualization, Methodology, Writing - original draft, Writing - review and editing; Jean-François Rupprecht, Formal analysis, Investigation, Writing - review and editing; Timothy E Saunders, Conceptualization, Resources, Formal analysis, Supervision, Funding acquisition, Visualization, Writing - original draft, Writing - review and editing

### Author ORCIDs

Anqi Huang (ID) https://orcid.org/0000-0003-0551-1160
Timothy E Saunders (ID) https://orcid.org/0000-0001-5755-0060

### Decision letter and Author response

Decision letter https://doi.org/10.7554/eLife.47380.sa1
Author response https://doi.org/10.7554/eLife.47380.sa2

## Additional files

### Supplementary files
• Transparent reporting form

## Data availability

All data generated or analysed during this study are included in the manuscript and supplementary files.

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
