## [Decision Letter]

Thank you for submitting your article "Embryonic geometry underlies phenotypic variation in decanalized conditions" for consideration by *eLife*. Your article has been reviewed by two peer reviewers, and the evaluation has been overseen by a Reviewing Editor and Naama Barkai as the Senior Editor. The reviewers have opted to remain anonymous.

The reviewers have discussed the reviews with one another and the Reviewing Editor has drafted this decision to help you prepare a revised submission.

Summary:

Huang and Saunders investigate phenotypic variability of bicoid mutant *Drosophila* embryos. Remarkably, they find that in mutant embryos, the number of engrailed stripes and the number of abdominal segments is determined by the length of the embryo. This rule applies to embryos with abnormally short AP axes generated by knockdown of fat2. In contrast, in the presence of Bcd, the correct number of segments always arise, regardless of embryo length, even in fat2 KD embryos. Thus, Bcd is required for scaling of expression patterns to accommodate variation in embryo length. Measurements of Bcd-GFP strongly suggests that this scaling is achieved by reducing the amplitude of the Bcd gradient concomitant with the reduction in egg length. By examining gap gene expression, they show that in short *bcd* mutant eggs, the distance between anterior and posterior poles must be sufficiently large to permit the expression of the genes Kruppel and knirps. Their data are consistent with a view of the bicoid mutant phenotype in which Torso signaling is not scaled in accord with embryo length, so that terminal signaling in short embryos overrides factor(s) that would otherwise permit the expression of Kr and knirps. Finally, the authors show that excess Bcd also disrupts patterning by interfering with specification in the vicinity of the fourth abdominal segment, with additional segments affected in shorter embryos.

The work is exciting because it articulates an explanation for phenotypic variability in *bcd* mutants. The manuscript is significant in showing that the Bcd gradient itself provides a buffer against variation in egg length.

Essential revisions:

The paper will benefit from re-structuring, to better explain the main findings. This is detailed in the individual reports, but in the discussion, the reviewers reached an agreement on what they find to be promising approach, which is described below. In addition, to fully establish the main findings, the following additional data are requested:

1) Please explore further how *bcd* mRNA and protein distributions are affected by fat2 KD, since this may help explain how *bcd* provides scaling. If the simulation can be used to address this point, the manuscript will be stronger. However, the authors must provide much more detail on how the simulation was implemented if the reader is to believe any conclusion drawn from it. As it stands now, the simulation is a distraction from the results.

2) Please perform an in situ hybridization and quantification of *bcd* mRNA, as suggested in the review

3) Please include an experimental verification of embryo shrinking.

Suggestion for revising the writing:

The opening question can remain the same ("what is canalization buffering against"); the authors show buffering against a wide range of embryo sizes, even well outside the normal range; this sets up the question of what AP patterning factors are required for canalization; then authors show the requirement of Bcd: without *bcd*, aspects of the resulting mutant phenotype (like the number of En stripes) now correlate with embryo size. I also suggest to keep the Bcd overexpression data at the end of the manuscript. Then at the very least, the authors should discuss ways / propose models to explain how Bcd might provide scaling, possibly through altered mRNA expression, localization, and/or translation, with references to Jun Ma's set of work on this very subject. The manuscript would be greatly strengthened if the authors could provide evidence, experimental and/or using their (currently rather poorly explained) model, that one or more of these processes are different in embryos of different sizes. This would help the field understand how the Bcd protein gradient is reshaped (as they claim) and in turn how this reshaping confers scaling.

Reviewer #1:

Executive summary: Overall, I believe this paper contains enough material to support and justify an interesting point using a clever method. However, I also find that, as written, its order of presentation does the narrative a great disservice. Below, I advocate for a rather serious reorganization of presentation (which incidentally should also make the paper shorter). But provided the presentation is modified, I do believe this paper can be made acceptable for *eLife* without additional experimental work (except for one small point – see moderate point #3).

Details:

Embryonic development in the fruit fly is said to exhibit "canalization", generating reproducible phenotypic outcomes despite inter-individual differences in genetic background and environmental conditions, within a certain range. Outside this range ("decanalized conditions"), individuals become sensitive to variations that would otherwise be neutral. In this manuscript, Huang & Saunders ask exactly what sources of variation canalization is buffering against. They argue that, in addition to genetic and environmental perturbations, another previously unidentified source of variation is the "naturally variable embryonic geometry". [But see "minor point 2"]

To address this question ("what sources of variation is canalization buffering against?"), the authors designed a clever way to modify geometry of the embryo, plausibly without changing very much else [but see "minor point 3"]. Using their clever tool, the authors can effectively increase the variability in embryo geometry – thus testing the ability of the developmental system to buffer against such perturbations.

In the absence of other complicating factors, the embryo demonstrates a truly remarkable capacity to deal with perturbation of its geometry. Frankly, to me this is perhaps the most striking finding reported here, but this is not what the authors focus on.

Now, in my mind, the way for the authors to make their point would be to apply a pressure towards decanalization of development, and observe that the ability of the embryo to buffer the geometric perturbations is reduced. With no other problems, embryos are basically normal (Figure 3—figure supplement 1, panels EFGH are frankly amazing and deserve to be in the main text.) With some perturbation (e.g. temperature, or increased dosage of Bcd like here) the embryos are able to buffer small variations of geometry but not big ones. With an even larger perturbation, embryos with non-normal aspect ratio fail to develop. This is the approach I would have expected, given the authors' stated goals.

But the authors instead apply a truly dramatic perturbation, removing maternal *bcd*. This is a lethal mutant. They then observe that sometimes the embryo is short, and everything is broken completely. At other times the embryo is longer and everything is broken partially. (A few structures do form in a way that resembles normal – but the entire first half of the embryo is still gone.) Thus, observed results subsection 2, the degree to which the embryo is broken is correlated with embryo length. And if we make the embryo's life even more difficult by compounding the already lethal *bcd* knockout with a further perturbation of geometry, things get even worse. (Results subsection 4).

I do not understand how we can make claims about what perturbations a system had evolved to buffer by studying it in a completely dysfunctional regime, which is by definition irrelevant for development. The authors demonstrated that in a family of lethal mutants shorter embryos are even more screwed up than longer ones. But all of them are lethal, at which point counting "just how badly lethal" seems irrelevant.

The authors then go on to apply another set of dramatic perturbations – rather than increasing Bcd a little, they overexpress it 6x or 10-fold, the latter again being lethal.

By this point I personally was not buying the story at all – but finally, the paper gets to Figure 5 and subsection “The gene network breaks at susceptible point in decanalized conditions” onwards, which finally turned everything around. I found that result very strong and interesting. But if I were a reader rather than a reviewer, I would not have made it that far.

My proposal: if the point made in the Introduction/Discussion section is indeed the intended focus of the narrative, I believe the paper could make it point much more persuasively, while also being shorter:

• Introduce the fat2RNAi trick that allows generating embryos with a much more varied geometry than normal, and explain how that offers a fantastic window into studying canalization

• Observe that, absent other perturbations, the development is capable of buffering that "geometric" variability. Move supplementary panels from Figure 3 supplement to the main text.

• Stress the system by increasing *bcd* dosage (4x, 6x), and proceed through the argument at the end of the Results section.

This is the narrative that actually makes the point set up in the Introduction, and this point is a strong one: the construct developed by the authors allows directly probing the ability of a system to buffer variations – in a very clean, controlled setting, of which I know few. It is a remarkable setting and beautiful result. Only after this point is made, can the response to *bcd* removal really be seen as further reinforcing the authors' point – not before, because such dramatic perturbation is way beyond anything relevant for normal development and therefore canalization; on its own, this line of evidence alone would be questionable. E.g. last paragraph of subsection “Embryonic length dictates segmentation gene pattern in the absence of *bcd*” – these discussions are useful for mapping out who activates who in a pathway, but entirely irrelevant for normal development and "what canalization evolved for", surely! I would in fact consider largely moving all the discussions of non-viable mutants to the supplement.

Other specific issues:

Moderate points:

1) Paragraph two of subsection “Embryonic length dictates segmentation gene pattern in the absence of *bcd*” unexpectedly bring up the question of expression pattern scaling and I'm not sure whether this is a good move. Exactly what is the paper contributing to that conversation? It seems like a distraction. I suggest removing.

2) "Genetic network breaks at susceptible point" – what does this sentence mean? How does a genetic network "break"? What is a "susceptible point"? Is this a result, or some intuition the authors are trying to communicate? The strongest point in this paper is introduced by the weakest/meaningless header. Similarly, when the Abstract is revised, I would suggest removing the last sentence referring to "vulnerable points in the network" which again sounds like some intuition (belonging to Discussion section) rather than a finding.

3) Second paragraph of subsection “Measurement of embryonic geometry”: I found this hard to parse (particularly the use of the term "further"). My interpretation is that EL pre-fixation is inferred from EL post-fixation assuming that aspect ratio is unchanged – correct? This should be clarified. But even more importantly – was this assumption of isotropic shrinkage verified? In my experience a harsh fixation protocol can lead to extensive deformations, their isotropic nature is not obvious, and given the role played by the aspect ratio in this paper, this seems an important point to demonstrate. [This is the one place where additional data may be required, but shouldn't be too onerous to obtain]

4) SDD simulation: I am very confused.

a) If I use an cylinder to approximate an ellipsoid, then geometrically, the "best approximation" cylinder is surely narrower than the embryo. Not wider. It seems like width is used as a free fitting parameter to make the data fit better, but the text tries to make it sound like this larger-than-actual radius was somehow expected, and I don't understand the argument.

b) Further, I believe the diffusion parameters, both in the bulk and at the surface, for a small molecule like Bcd or Hb, were previously measured in the embryo specifically. So it sounds like this could be a zero-parameter fit. How bad is it? Why the discrepancy?

c) And in particular, if using an elliptical geometry truly does not significantly alter the results, why not use it, given that the parameters of the actual geometry are, again, known?

d) Does the SDD simulation also reproduce the features that the plot of Figure 3G specifically highlights, namely the larger concentration in the posterior and the crossing point? The panel D in Figure 3 supplement does not allow to see this.

Reviewer #2:

Huang and Saunders investigate phenotypic variability of bicoid mutant *Drosophila* embryos. Remarkably, they find that in mutant embryos, the number of engrailed stripes and the number of abdominal segments is determined by the length of the embryo. This rule applies to embryos with abnormally short AP axes generated by knockdown of fat2. In contrast, in the presence of Bcd, the correct number of segments always arise, regardless of embryo length, even in fat2 KD embryos. Thus, Bcd is required for scaling of expression patterns to accommodate variation in embryo length. Measurements of Bcd-GFP strongly suggests that this scaling is achieved by reducing the amplitude of the Bcd gradient concomitant with the reduction in egg length. By examining gap gene expression, they show that in short *bcd* mutant eggs, the distance between anterior and posterior poles must be sufficiently large to permit the expression of the genes Kruppel and knirps. Their data are consistent with a view of the bicoid mutant phenotype in which Torso signaling is not scaled in accord with embryo length, so that terminal signaling in short embryos overrides factor(s) that would otherwise permit the expression of Kr and knirps. Finally, the authors show that excess Bcd also disrupts patterning by interfering with specification in the vicinity of the fourth abdominal segment, with additional segments affected in shorter embryos.

The work is exciting because it articulates an explanation for phenotypic variability in *bcd* mutants. The manuscript is significant in showing that the Bcd gradient itself provides a buffer against variation in egg length. The main finding, at least to this reviewer, is that Bcd itself is required for length scaling. The authors should draw greater attention to this result. This result seems consistent with the work of Jun Ma, whose publications should be cited (e.g., doi: 10.1242/dev.064402). Interestingly, scaling occurs in fat2-depleted embryos despite the paradoxical finding of more *bcd* mRNA upon fat2 KD (Figure 5—figure supplement 1). This suggests alteration in Bcd translation and/or localization of mRNA in short eggs. The manuscript would be strengthened if the authors could comment further about the nature of Bcd-mediated canalization, particularly to explain why more mRNA generates less protein. Along these lines, more detailed description of the simulation would be helpful. In particular, it would strengthen the manuscript to know whether *bcd* mRNA localization might play a role in canalization. What geometry of *bcd* mRNA was used in the simulation? It should not be beyond the technical limits of the lab to perform a semi-quantitative in situ hybridization for *bcd* mRNA to address the question of whether *bcd* mRNA distribution is changed by fat2 KD in a way that helps explain Bcd-mediated scaling.

As a final major comment, the authors document variability in gap gene boundaries in control embryos, fat2 KD, and *bcd* mutants (Figure 1 J-M, Figure 1—figure supplement 1, and Figure 3—figure supplement 1). The authors should address how much of this variability arises from variability in egg length. The LlamaTag experiment suggests that at some of the variability originates from differences in egg length (this is clear from the anticorrelation in Figure 3I). If egg length determines boundary positions, then there should be a relationship between egg length and gap gene boundary position in *bcd* mutants. The degree of correlation might be even stronger in *bcd* mutants due to the absence of the canalizing activity of Bcd. The authors should plot boundary position for each gap gene boundary as a function of egg length, similar to Figure 3I.

[Editors' note: further revisions were suggested prior to acceptance, as described below.]

Thank you for resubmitting your work entitled "Embryonic geometry underlies phenotypic variation in decanalized conditions" for further consideration by *eLife*. Your revised article has been evaluated by Naama Barkai (Senior Editor) and a Reviewing Editor.

The manuscript has been improved but there are some remaining issues that need to be addressed before acceptance. please address the remaining comments, as specified below

Reviewer #1:

I believe the revisions improved the manuscript significantly.

Reviewer #2:

The revised manuscript will be satisfactory for publication if the authors can clarify the following points listed below regarding the simulation of Bcd-GFP production. The main point of Figure 1F-K is to address how the Bcd gradient changes in embryos with altered geometry. My concern is that the presentation of the simulation in the text may appear at first glance incongruent with the figures. In the end I believe the figures are consistent with the text, but I would like some clarification. I imagine what I am requesting will be trivial for the authors to provide, but at the same time very useful and insightful for readers.

1) The authors state that the same parameter values describe the distribution of fluorescent Bcd-GFP regardless of embryo geometry. Unless I am mistaken, this means that, at steady-state imposed by the model (and using the finding that *bcd* mRNA doesn't change very much in terms of its total amount), the total amount of Bcd-GFP embryo-wide is equivalent in all embryos regardless of their geometry. Further, the total amounts of fluorescent and non-fluorescent Bcd-GFP are also equivalent. It would be helpful to explicitly state this in the text for those not immediately familiar with modeling and/or may not be aware that the maturation time of GFP is non-negligible in this context.

2) Assuming that Bcd-GFP amount is the same in all embryos, a reader will then examine Figure 1G and note that the summed nuclear intensity (or in some sense, the area under the curve) does not appear to be the same in embryos of different sizes. Instead it appears that Bcd-GFP scales with embryo length. This appears to contradict the idea that total BcdGFP is unchanged. However, the authors say that this is due to "dilution" of Bcd-GFP. I interpret this to mean that if one considers an embryo as a set of "salami slices," then, due to the altered aspect ratio, a slice at some absolute distance from the anterior pole in fat2 KD (or, more generally, in short) embryos has a larger volume compared to a slice at the same distance in wild-type. The apparent reduction in "nuclear concentration" of Bcd-GFP occurs because whatever amount of Bcd-GFP is available at that distance is roughly the same between long and short embryos. This amount distributes into the available volume, so that the apparent concentration of BcdGFP appears lower in fat2 KD. Could the authors please state this more explicitly, if my interpretation is correct. I believe that splitting the long sentence the fifth paragraph of subsection “Developmental reproducibility is preserved with a minor impact on scaling under geometrical perturbations” into two or three sentences will suffice, devoting a sentence or two to each of the two differences (low anterior, high posterior) in the gradients of long and short eggs. This will help the reader to understand why the 1D plot of Bcd-GFP vs AP does not accurately represent the total amount of Bcd and results in the reduction in nuclear intensity in the anterior.

3) Figure 1 presents data and simulation regarding only fluorescent Bcd-GFP. A powerful aspect of the simulation is that it can estimate the quantity that we (and the embryo) actually care about, the distribution of _total_ Bcd. Given the sizable maturation time of 50 minutes used in the simulation, my intuition is that the invisible proportion likely extends over a significantly larger fraction of egg length in short embryos compared to long. However my intuition may well be wrong. Could the authors please present and discuss the estimated _total_ Bcd-GFP distribution as a function of both absolute and normalized distance in embryos of various aspect ratios (lengths)? To my thinking, this is the only way to address the question of how the actual gradient (not just the visible one) changes with changing geometry. For example, after estimating the gradients of total Bcd, could the authors now show the difference in total Bcd concentration, as in Figure 1K lower panel, except now for total Bcd. (Could the authors also include error bars on the difference in 1K and in a new plot regarding total concentration?) Moreover, the difference in total Bcd can be converted into a difference in fractional egg length ("positional error") as a function of normalized AP position, i.e., the difference in the positions at which the same Bcd concentration is found in short eggs compared to long. This difference is zero near the midpoint of the AP axis where the curves cross and becomes larger moving toward either pole. This will give the reader a sense of magnitude of the mistakes in assigning positional identity that we would expect in short embryos if Bcd were the only patterning cue. In turn, these differences can be compared to the changes in the placement of gap gene expression boundaries presented in Figure 2. Are the changes in the Bcd gradient predictive of the changes in boundary placement (in other words, do gap gene boundaries arise at the same Bcd concentration in long and short eggs)? Do gap gene boundaries shift more, less, or the same amount as predicted by the changes in the Bcd gradient? The answers will help address how the altered Bcd gradient in short embryos generates (or works against!) scaling, and how much of the scaling phenomenon must arise independently of the Bcd gradient.

---

## [Author Response]

Essential revisions:The paper will benefit from re-structuring, to better explain the main findings. This is detailed in the individual reports, but in the discussion, the reviewers reached an agreement on what they find to be promising approach, which is described below.

We agree that the structure of the original submission was not clear. We have now reworked presentation of the data such that: (1) we first characterize the *fat2RNAi* embryos regarding the problem of scaling and reproducibility of segmentation gene expression; (2) then we characterize the effects of embryo geometry under the decanalized condition of Bcd over-expression; (3) and finally we characterize the effects of embryo geometry under the decanalized condition of *bcd* mutant. We outline specific changes below in our detailed response to specific points raised.

In addition, to fully establish the main findings, the following additional data are requested:1) Please explore further how bcd mRNA and protein distributions are affected by fat2 KD, since this may help explain how bcd provides scaling. If the simulation can be used to address this point, the manuscript will be stronger. However, the authors must provide much more detail on how the simulation was implemented if the reader is to believe any conclusion drawn from it. As it stands now, the simulation is a distraction from the results.

We have carried out fluorescent in situ hybridization against *bcd* mRNA in both wild type and *fat2 KD* embryos. We found that, on an absolute scale, the *bcd* mRNA in *fat2 KD* distributes to a similar extent compared to that of the wild type. We have presented these data in Figure 1I-J and subsection “Developmental reproducibility is preserved with a minor impact on scaling under geometrical perturbations”. Further, we have included better simulations that use more realistic embryo geometry. This new work was done with a theorist, Jean-Francois Rupprecht, who performed the new simulations in more realistic geometries. We outline specific changes below in our detailed response to specific points raised.

2) Please perform an in situ hybridization and quantification of bcd mRNA, as suggested in the review

We have performed these experiments. We outline specific changes below. Briefly, these experiments (a) validate that the 10x *bcd* line significantly increases the levels of *bcd* mRNA present; and (b) the *fat2RNAi* line does not have noticeably altered *bcd* mRNA levels compared to wild-type embryos. This is consistent with a previously reported volume dependence on *bcd* mRNA levels (Cheung et al., 2011), as our *fat2RNAi* embryos do not have substantially reduced embryo volume.

3) Please include an experimental verification of embryo shrinking.

We have now performed additional quantifications to test the impact of fixation on the embryo geometry. We do not observe major shifts in the aspect ratio of the embryos before and after fixation, supporting the assumption of isotropic embryo shrinkage. We now present this data in Figure 1—figure supplement 1B-C. We discuss these results further below.

Suggestion for revising the writing:The opening question can remain the same ("what is canalization buffering against"); the authors show buffering against a wide range of embryo sizes, even well outside the normal range; this sets up the question of what AP patterning factors are required for canalization; then authors show the requirement of Bcd: without bcd, aspects of the resulting mutant phenotype (like the number of En stripes) now correlate with embryo size. I also suggest to keep the Bcd overexpression data at the end of the manuscript. Then at the very least, the authors should discuss ways / propose models to explain how Bcd might provide scaling, possibly through altered mRNA expression, localization, and/or translation, with references to Jun Ma's set of work on this very subject. The manuscript would be greatly strengthened if the authors could provide evidence, experimental and/or using their (currently rather poorly explained) model, that one or more of these processes are different in embryos of different sizes. This would help the field understand how the Bcd protein gradient is reshaped (as they claim) and in turn how this reshaping confers scaling.

We thank the reviewers for detailing their suggested revised structure. We broadly agree with the above reworking. We now detail the behaviour of the *fat2RNAi* embryos first under normal Bcd expression levels. We describe in more depth the effects of embryo size change on Bcd and the gap genes. We then introduce the effects of geometry on patterning when Bcd is over-expressed. We finish with discussing the effects of geometry on patterning when Bcd is depleted. We believe the last section is important – important mechanistic insights can be made from mutant embryos, and we clarify the point in the text.

Reviewer #1:Executive summary: Overall, I believe this paper contains enough material to support and justify an interesting point using a clever method. However, I also find that, as written, its order of presentation does the narrative a great disservice. Below, I advocate for a rather serious reorganization of presentation (which incidentally should also make the paper shorter). But provided the presentation is modified, I do believe this paper can be made acceptable for eLife without additional experimental work (except for one small point – see moderate point #3).

With regard to writing, please see the above replies.

Details:Embryonic development in the fruit fly is said to exhibit "canalization", generating reproducible phenotypic outcomes despite inter-individual differences in genetic background and environmental conditions, within a certain range. Outside this range ("decanalized conditions"), individuals become sensitive to variations that would otherwise be neutral. In this manuscript, Huang & Saunders ask exactly what sources of variation canalization is buffering against. They argue that, in addition to genetic and environmental perturbations, another previously unidentified source of variation is the "naturally variable embryonic geometry". [But see "minor point 2"]To address this question ("what sources of variation is canalization buffering against?"), the authors designed a clever way to modify geometry of the embryo, plausibly without changing very much else [but see "minor point 3"]. Using their clever tool, the authors can effectively increase the variability in embryo geometry – thus testing the ability of the developmental system to buffer against such perturbations.In the absence of other complicating factors, the embryo demonstrates a truly remarkable capacity to deal with perturbation of its geometry. Frankly, to me this is perhaps the most striking finding reported here, but this is not what the authors focus on.

We agree this observation is truly remarkable. We have restructured the paper such that this result is now apparent in Figures 1-2 of the revised manuscript.

Now, in my mind, the way for the authors to make their point would be to apply a pressure towards decanalization of development, and observe that the ability of the embryo to buffer the geometric perturbations is reduced. With no other problems, embryos are basically normal (Figure 3—figure supplement 1, panels EFGH are frankly amazing and deserve to be in the main text.)

We agree with this suggestion. We have restructured the manuscript and moved the suggested images to revised Figure 2.

With some perturbation (e.g. temperature, or increased dosage of Bcd like here) the embryos are able to buffer small variations of geometry but not big ones. With an even larger perturbation, embryos with non-normal aspect ratio fail to develop. This is the approach I would have expected, given the authors' stated goals.But the authors instead apply a truly dramatic perturbation, removing maternal bcd. This is a lethal mutant. They then observe that sometimes the embryo is short, and everything is broken completely. At other times the embryo is longer and everything is broken partially. (A few structures do form in a way that resembles normal – but the entire first half of the embryo is still gone.) Thus, observed results subsection 2, the degree to which the embryo is broken is correlated with embryo length. And if we make the embryo's life even more difficult by compounding the already lethal bcd knockout with a further perturbation of geometry, things get even worse. (Results subsection 4).I do not understand how we can make claims about what perturbations a system had evolved to buffer by studying it in a completely dysfunctional regime, which is by definition irrelevant for development. The authors demonstrated that in a family of lethal mutants shorter embryos are even more screwed up than longer ones. But all of them are lethal, at which point counting "just how badly lethal" seems irrelevant.

The use of mutant embryos to understand gene interactions has been a very powerful tool, especially in *Drosophila.* As the reviewer highlights, a *bcd* null mutant is a severe perturbation, yet we still observe correlation in phenotype with embryo length. The large variability in *bcd* null mutant phenotypes has been discussed previously, but crucially, here, we demonstrate that this phenotypic variability is not random. Instead, the severity of phenotypes directly correlates with embryo length. This suggests that the ability of the gap gene network – which is largely still there in the absence of Bcd – to scale to embryo size is lost under Bcd depletion.

Our results explain a longstanding problem, noticed in papers from the 1980’s: there is large phenotypic variability in *bcd* null mutants, and this variability is, at least partially, explainable by embryo geometry. This result may well be applicable to other systems, and therefore we believe it should be made clearly in the manuscript. We discuss this point in subsection “AP Patterning of bcd mutants correlates with embryonic length” and in the Discussion.

The authors then go on to apply another set of dramatic perturbations – rather than increasing Bcd a little, they overexpress it 6x or 10-fold, the latter again being lethal.

The *Drosophila* embryo is known to be remarkably robust to increases in Bcd dosage. Namba et al.,1997 reported that embryos developed into larvae even with 6x- Bcd levels. Here, we apply 4x, 6x and 10x perturbations. Given the robustness of the system, we can characterize these as mild, middle, and strong perturbations respectively. Therefore, we are exploring a reasonable range of perturbation. For example, in revised Figure 3E we clearly show the relationship between Bcd overexpression and embryo geometry on pattering outcomes.

By this point I personally was not buying the story at all – but finally, the paper gets to Figure 5 and subsection “The gene network breaks at susceptible point in decanalized conditions” onwards, which finally turned everything around. I found that result very strong and interesting. But if I were a reader rather than a reviewer, I would not have made it that far.

We have improved the discussion of the above points to make their scientific importance more clear. Secondly, we have moved the material from Figure 5 into revised Figures 3 and 4. We have given greater prominence to these results and explained their significance in more detail.

My proposal: if the point made in the Introduction/Discussion section is indeed the intended focus of the narrative, I believe the paper could make it point much more persuasively, while also being shorter:• Introduce the fat2RNAi trick that allows generating embryos with a much more varied geometry than normal, and explain how that offers a fantastic window into studying canalization

We have reordered the manuscript to make the *fat2RNAi* results with normal Bcd levels more obvious – this now constitutes revised Figures 1 and 2.

• Observe that, absent other perturbations, the development is capable of buffering that "geometric" variability. Move supplementary panels from Figure 3 supplement to the main text.

We have included this data in revised Figure 2.

• Stress the system by increasing bcd dosage (4x, 6x), and proceed through the argument at the end of the Results section.

We have now brought these results forward, and they form the basis of revised Figures 3 and 4.

This is the narrative that actually makes the point set up in the Introduction, and this point is a strong one: the construct developed by the authors allows directly probing the ability of a system to buffer variations – in a very clean, controlled setting, of which I know few. It is a remarkable setting and beautiful result. Only after this point is made, can the response to bcd removal really be seen as further reinforcing the authors' point – not before, because such dramatic perturbation is way beyond anything relevant for normal development and therefore canalization; on its own, this line of evidence alone would be questionable. E.g. last paragraph of subsection “Embryonic length dictates segmentation gene pattern in the absence of bcd” – these discussions are useful for mapping out who activates who in a pathway, but entirely irrelevant for normal development and "what canalization evolved for", surely! I would in fact consider largely moving all the discussions of non-viable mutants to the supplement.

We understand the reviewers point, but we do feel that the depleted Bcd condition provides important insights that cannot be gleaned from the over-expression mutants. For example, as stated above, these results provide an important insight into how the large phenotypic variability often seen in mutant conditions of key patterning genes may have (at least to some degree) a systematic cause related to embryo size. In the revised manuscript, these results now constitute revised Figures 5 and 6.

Other specific issues:Moderate points:1) Paragraph two of subsection “Embryonic length dictates segmentation gene pattern in the absence of bcd” unexpectedly bring up the question of expression pattern scaling and I'm not sure whether this is a good move. Exactly what is the paper contributing to that conversation? It seems like a distraction. I suggest removing.

We agree that where this point was raised in the original manuscript was confusing. We do feel that an important point here is being discussed regarding how scaling occurs. Therefore, we have moved this part in to the Discussion and reworked it to highlight the key points.

2) "Genetic network breaks at susceptible point" – what does this sentence mean? How does a genetic network "break"? What is a "susceptible point"? Is this a result, or some intuition the authors are trying to communicate? The strongest point in this paper is introduced by the weakest/meaningless header. Similarly, when the Abstract is revised, I would suggest removing the last sentence referring to "vulnerable points in the network" which again sounds like some intuition (belonging to Discussion section) rather than a finding.

We have reworked this section, including the header. By “susceptible”, we meant the point in the patterning network where we most often see patterning defects under perturbation (either to embryo size or Bcd expression levels, or both). However, we agree that the original wording was not clear. As suggested, we have removed these words from the Abstract and Results, and elaborated these concepts in the Discussion.

3) Second paragraph of subsection “Measurement of embryonic geometry”: I found this hard to parse (particularly the use of the term "further"). My interpretation is that EL pre-fixation is inferred from EL post-fixation assuming that aspect ratio is unchanged – correct? This should be clarified. But even more importantly – was this assumption of isotropic shrinkage verified? In my experience a harsh fixation protocol can lead to extensive deformations, their isotropic nature is not obvious, and given the role played by the aspect ratio in this paper, this seems an important point to demonstrate. [This is the one place where additional data may be required, but shouldn't be too onerous to obtain]

We thank the reviewer for raising this important point which was insufficiently explained in the original submission. As described in the Materials and methods section, we fix the embryos in heptane saturated by 37% paraformaldehyde for 1 hr, after which the embryos are devitellinized manually. As this fixation method does not involve any vortex or heat shock step, it generally preserves embryo morphology. The fixed embryos only shrink during mounting, when the mounting medium is added (Aqua-Poly/Mount).

In the revised manuscript, we have carried out quantifications to verify that such shrinkage is close to isotropic by nature. We measured the length and width of OreR and *fat2RNAi* embryos, in both live and fixed conditions (n = 239 each condition). We found that the aspect ratio of the OreR embryos remains comparable prior and post fixation (Figure 1—figure supplement 1B-C). This suggests isotropic shrinkage and supports our approach to use aspect ratio of post-fixation embryos to estimate their original embryonic length.

We do see a smaller aspect ratio of *fat2RNAi* embryos post fixation. However, this can be explained by the differential collection of embryos. While the live condition measurements are carried out on a stereoscope, where we collected *fat2RNAi* embryos spanning all geometrical range, the fix condition measurements are obtained using confocal staining data, where we focused on imaging *fat2RNAi* individuals of extremely short length.

4) SDD simulation: I am very confused.a) If I use an cylinder to approximate an ellipsoid, then geometrically, the "best approximation" cylinder is surely narrower than the embryo. Not wider. It seems like width is used as a free fitting parameter to make the data fit better, but the text tries to make it sound like this larger-than-actual radius was somehow expected, and I don't understand the argument.

In the original submission we tried to keep the model as simple as possible, while including 3D effects of embryo size. However, it is clear that our approximation was unhelpful. In the revised manuscript we now perform our simulations on the surface of an ellipsoid – a much better approximation to the embryo geometry.

b) Further, I believe the diffusion parameters, both in the bulk and at the surface, for a small molecule like Bcd or Hb, were previously measured in the embryo specifically. So it sounds like this could be a zero-parameter fit. How bad is it? Why the discrepancy?

The reviewer is correct; effectively this is a zero parameter fit. Taking the measured diffusion coefficient and degradation time from Durrieu et al.,2018 and accounting for the eGFP folding time (~50 minutes), we are able to fit reasonably well both the Bcd concentration profile in n.c. 13 for both wild-type and *fat2RNAi* embryos by simply altering the embryo geometry and no other parameters.

We now present this data more clearly in Figure 1, making assessment of fit quality easier.

c) And in particular, if using an elliptical geometry truly does not significantly alter the results, why not use it, given that the parameters of the actual geometry are, again, known?

We have now included the full elliptical solution.

d) Does the SDD simulation also reproduce the features that the plot of Figure 3G specifically highlights, namely the larger concentration in the posterior and the crossing point? The panel D in Figure 3 supplement does not allow to see this.

We still observe the crossing point in concentrations between wild type and *fat2RNAi* embryos (when the AP axis is scaled), and the higher Bcd concentration in the posterior for *fat2RNAi* embryos. In the revised Figure 1 this is clearer.

Reviewer #2:Huang and Saunders investigate phenotypic variability of bicoid mutant *Drosophila* embryos. Remarkably, they find that in mutant embryos, the number of engrailed stripes and the number of abdominal segments is determined by the length of the embryo. This rule applies to embryos with abnormally short AP axes generated by knockdown of fat2. In contrast, in the presence of Bcd, the correct number of segments always arise, regardless of embryo length, even in fat2 KD embryos. Thus, Bcd is required for scaling of expression patterns to accommodate variation in embryo length. Measurements of Bcd-GFP strongly suggests that this scaling is achieved by reducing the amplitude of the Bcd gradient concomitant with the reduction in egg length. By examining gap gene expression, they show that in short bcd mutant eggs, the distance between anterior and posterior poles must be sufficiently large to permit the expression of the genes Kruppel and knirps. Their data are consistent with a view of the bicoid mutant phenotype in which Torso signaling is not scaled in accord with embryo length, so that terminal signaling in short embryos overrides factor(s) that would otherwise permit the expression of Kr and knirps. Finally, the authors show that excess Bcd also disrupts patterning by interfering with specification in the vicinity of the fourth abdominal segment, with additional segments affected in shorter embryos.The work is exciting because it articulates an explanation for phenotypic variability in bcd mutants. The manuscript is significant in showing that the Bcd gradient itself provides a buffer against variation in egg length. The main finding, at least to this reviewer, is that Bcd itself is required for length scaling. The authors should draw greater attention to this result. This result seems consistent with the work of Jun Ma, whose publications should be cited (e.g., doi: 10.1242/dev.064402).

Our results are consistent with previous studies (Cheung et al., 2011, 2014) showing that the scaling of embryonic patterns can be traced back to the scaling of the Bcd gradient itself. Although the phenomena are similar, the mechanisms underlying the adaptive Bcd gradient formation are different. This is because the macroscopic variable that the patterning system is buffering against is different (volume and geometry, respectively). Embryos adapt to volume variation by altering the amount and distribution of mRNA deposition (Cheung et al., 2011). In our case, geometry changes but not (substantially) volume. In our case, we found that SDD model is sufficient to generate different Bcd profiles in embryos of different geometry without altering any other dynamic parameters. We have improved discussion of this point, and it is included within Figure 1.

Interestingly, scaling occurs in fat2-depleted embryos despite the paradoxical finding of more bcd mRNA upon fat2 KD (Figure 5—figure supplement 1). This suggests alteration in Bcd translation and/or localization of mRNA in short eggs.

In our new FISH data, we do not see a significant change in the distribution of *bcd* mRNA between wild-type and *fat2RNAi* embryos. Though it is challenging to quantify FISH data, our results do not reveal an obvious change in the *bcd* mRNA levels in *fat2RNAi* embryos.

Further, in the revised manuscript we are able to fit the differing Bcd profiles between wild-type and *fat2RNAi* embryos by simply accounting for embryo geometry – *i.e.* no change in production, diffusion or degradation of Bcd. Therefore, our evidence currently suggests that the translation and/or localization of Bcd is not significantly altered in *fat2RNAi* embryos.

These results are broadly consistent with Cheung et al.,2011. In that paper, they show that Bcd concentration is related to embryo volume. In our case, even the shortest embryos are still ~90% of the volume of wild-type embryos (assuming ellipsoidal geometry). Therefore, our observed stability of the Bcd production in the different embryos is consistent.

The manuscript would be strengthened if the authors could comment further about the nature of Bcd-mediated canalization, particularly to explain why more mRNA generates less protein.

See answer above and also below regarding the new FISH data.

Along these lines, more detailed description of the simulation would be helpful. In particular, it would strengthen the manuscript to know whether bcd mRNA localization might play a role in canalization. What geometry of bcd mRNA was used in the simulation?

We have improved the simulations. We now use an ellipsoidal geometry. From the FISH data, we do not see a significant shift in the *bcd* mRNA profile. Therefore, in the simulations we use the same production domain for both wild-type and *fat2RNAi* embryos.

It should not be beyond the technical limits of the lab to perform a semi-quantitative in situ hybridization for bcd mRNA to address the question of whether bcd mRNA distribution is changed by fat2 KD in a way that helps explain Bcd-mediated scaling.

We agree with the Reviewer and have undertaken this experiment. The fixation protocol required optimization, which took a few months. We performed FISH on wild-type, *fat2RNAi* and *10*x-*bcd* embryos. In the latter case, we see a clear increase in both the magnitude and extent of *bcd* mRNA. For the other two scenarios the profile of *bcd* mRNA is comparable.

As the Reviewer rightfully highlights, this analysis is only semi-quantitative. To compare embryos, we normalized signal by the background signal apparent in embryos ~120μm from the embryo anterior pole. We cannot definitively state that the wild-type and *fat2RNAi* embryos have the same *bcd* mRNA profiles. However, it is unlikely that any variations are large; which contrasts with the 10x-*bcd* line, which has clear change in both magnitude and profile extent.

This data is now included in Figure 1, Figure 1—figure supplement 1 and Figure 3—figure supplement 1.

As a final major comment, the authors document variability in gap gene boundaries in control embryos, fat2 KD, and bcd mutants (Figure 1 J-M, Figure 1—figure supplement 1, and Figure 3—figure supplement 1). The authors should address how much of this variability arises from variability in egg length. The LlamaTag experiment suggests that at some of the variability originates from differences in egg length (this is clear from the anticorrelation in Figure 3I). If egg length determines boundary positions, then there should be a relationship between egg length and gap gene boundary position in bcd mutants. The degree of correlation might be even stronger in bcd mutants due to the absence of the canalizing activity of Bcd. The authors should plot boundary position for each gap gene boundary as a function of egg length, similar to Figure 3I.

We have plotted the relative boundary positions against embryonic length for all four gap genes in *bcdKO* and *bcdKO, fat2RNAi* embryos (Figure 6—figure supplement 2). Most of the relative boundary positions show strong correlation with embryonic length, demonstrating the loss of scaling.

[Editors' note: further revisions were suggested prior to acceptance, as described below.]

Reviewer #2:The revised manuscript will be satisfactory for publication if the authors can clarify the following points listed below regarding the simulation of Bcd-GFP production. The main point of Figure 1F-K is to address how the Bcd gradient changes in embryos with altered geometry. My concern is that the presentation of the simulation in the text may appear at first glance incongruent with the figures. In the end I believe the figures are consistent with the text, but I would like some clarification. I imagine what I am requesting will be trivial for the authors to provide, but at the same time very useful and insightful for readers.1) The authors state that the same parameter values describe the distribution of fluorescent Bcd-GFP regardless of embryo geometry. Unless I am mistaken, this means that, at steady-state imposed by the model (and using the finding that bcd mRNA doesn't change very much in terms of its total amount), the total amount of Bcd-GFP embryo-wide is equivalent in all embryos regardless of their geometry. Further, the total amounts of fluorescent and non-fluorescent Bcd-GFP are also equivalent. It would be helpful to explicitly state this in the text for those not immediately familiar with modeling and/or may not be aware that the maturation time of GFP is non-negligible in this context.

Yes, the reviewer interpretation is correct. Because the *fat2RNAi* embryos are bigger in embryonic width than wild-type embryos, the change in volume is not large despite the change in the anterior-posterior length. Therefore, from our model, the total amount of Bcd in the embryos does not change substantially. We have clarified this in the text during discussion of the modelling (subsection “Developmental reproducibility is preserved with a minor impact on scaling under geometrical perturbations”).

2) Assuming that Bcd-GFP amount is the same in all embryos, a reader will then examine Figure 1G and note that the summed nuclear intensity (or in some sense, the area under the curve) does not appear to be the same in embryos of different sizes. Instead it appears that Bcd-GFP scales with embryo length. This appears to contradict the idea that total BcdGFP is unchanged. However, the authors say that this is due to "dilution" of Bcd-GFP. I interpret this to mean that if one considers an embryo as a set of "salami slices," then, due to the altered aspect ratio, a slice at some absolute distance from the anterior pole in fat2 KD (or, more generally, in short) embryos has a larger volume compared to a slice at the same distance in wild-type. The apparent reduction in "nuclear concentration" of Bcd-GFP occurs because whatever amount of Bcd-GFP is available at that distance is roughly the same between long and short embryos. This amount distributes into the available volume, so that the apparent concentration of BcdGFP appears lower in fat2 KD. Could the authors please state this more explicitly, if my interpretation is correct. I believe that splitting the long sentence the fifth paragraph of subsection “Developmental reproducibility is preserved with a minor impact on scaling under geometrical perturbations” into two or three sentences will suffice, devoting a sentence or two to each of the two differences (low anterior, high posterior) in the gradients of long and short eggs. This will help the reader to understand why the 1D plot of Bcd-GFP vs AP does not accurately represent the total amount of Bcd and results in the reduction in nuclear intensity in the anterior.

Yes, the reviewer interpretation is correct. We appreciate the advice on improving the description and we have clarified in the text (subsection “Developmental reproducibility is preserved with a minor impact on scaling under geometrical perturbations”).

3) Figure 1 presents data and simulation regarding only fluorescent Bcd-GFP. A powerful aspect of the simulation is that it can estimate the quantity that we (and the embryo) actually care about, the distribution of _total_ Bcd. Given the sizable maturation time of 50 minutes used in the simulation, my intuition is that the invisible proportion likely extends over a significantly larger fraction of egg length in short embryos compared to long. However my intuition may well be wrong. Could the authors please present and discuss the estimated _total_ Bcd-GFP distribution as a function of both absolute and normalized distance in embryos of various aspect ratios (lengths)? To my thinking, this is the only way to address the question of how the actual gradient (not just the visible one) changes with changing geometry.

We agree that the relevant readout biologically is the local concentration of mature Bcd protein regardless of the folding state of the tagged GFP. However, we do not have data on the folding time of Bcd protein itself and so it is not straightforward to infer the functional Bcd concentration from the experimental data. In the modelling we now test two different scenarios. First, we assume Bcd folds at the same rate as eGFP, as shown in Figure 1. We also include in Figure 1—figure supplement 1G, the total nuclear Bcd concentration (assuming immediate folding of Bcd) as a function of absolute or normalized embryonic length. The real functional Bcd gradient lies between these two curves.

Interestingly, the profiles of long and short embryos intersect at a relatively more anterior position when we consider the total Bcd concentration (disregarding protein folding time). Intersection of the profiles now occurs around 25% EL vs. ~40% EL in our observed experimental profiles (which are affected by protein folding rates). To double-check our simulations, we used a simpler cylindrical geometry where we could analytically solve the SDD model to further validate this observation. Taking diffusion to only occur on the body of cylinders of equal surface area (ignoring ends) but different lengths (350um and 500um), we also see an anterior shift in the crossover position when considering the total Bcd concentration, Author response image 1. We discuss these results in paragraph five of subsection “Developmental reproducibility is preserved with a minor impact on scaling under geometrical perturbations”.

**Author response image 1. respfig1:** Bcd gradient at steady-state using SDD model on surface of a cylinder (blue: L = 500μm, r = 100μm; red: L =350μm, r = 143μm). Bcd is introduced at rate J at the anterior end of the cylinder. We exclude the ends of the cylinder. (**A**) Bcd gradient assuming immediate protein folding (total Bcd). (**B**) Bcd gradient accounting for protein folding time. Dashed lines highlight how the crossover point shifts due to accounting for folding..

For example, after estimating the gradients of total Bcd, could the authors now show the difference in total Bcd concentration, as in Figure 1K lower panel, except now for total Bcd. (Could the authors also include error bars on the difference in 1K and in a new plot regarding total concentration?)

We have replotted Figure 1K to include the experimental data with error bars. We have included a similar plot for the theory in Figure 1—figure supplement 1G. In summary, the parameter of protein folding time does not affect the relative shape of the gradient in long vs short embryos but does alter the relative position where curves intersect.

Moreover, the difference in total Bcd can be converted into a difference in fractional egg length ("positional error") as a function of normalized AP position, i.e., the difference in the positions at which the same Bcd concentration is found in short eggs compared to long. This difference is zero near the midpoint of the AP axis where the curves cross and becomes larger moving toward either pole. This will give the reader a sense of magnitude of the mistakes in assigning positional identity that we would expect in short embryos if Bcd were the only patterning cue. In turn, these differences can be compared to the changes in the placement of gap gene expression boundaries presented in Figure 2. Are the changes in the Bcd gradient predictive of the changes in boundary placement (in other words, do gap gene boundaries arise at the same Bcd concentration in long and short eggs)? Do gap gene boundaries shift more, less, or the same amount as predicted by the changes in the Bcd gradient? The answers will help address how the altered Bcd gradient in short embryos generates (or works against!) scaling, and how much of the scaling phenomenon must arise independently of the Bcd gradient.

This is a challenging question to answer. As shown by a number of labs, the positional information transfer from the Bcd gradient to gap gene expression is a dynamic and history dependent process. Liu et al.,2013, have shown that the gap gene boundary positions only correlate with absolute local Bcd concentration at early stages (10 min into n.c. 14), while in late stages (50 min into n.c. 14) the boundary positions depend on combinatorial inputs from maternal gradients and the gap genes themselves. Here Liu et al.perturb the Bcd gradient by altering maternal dosage, which changes mainly the amplitude but not the characteristic length of the gradient. However, how gap gene expression temporally integrates a gradient of different characteristic length (as in this study) remain unclear. Our results (Figure 2) only demonstrate that at late stages, the gap gene boundaries largely recapitulate the wild-type scenario, even with an altered Bcd gradient due to geometrical perturbation. But, providing precise predictions for the gap gene shifts due to Bcd perturbation is difficult without a full model of gap gene formation. We have in fact tried tackling this problem using the model published by Bieler et al., 2011. However, there are many biological and technical challenges in fully modelling such interactions and this represents a new piece of work. Indeed, we have had discussions with modelling groups, such as Manu, to explore this question further. We have added a sentence to the Discussion to highlight this interesting point.